

**Grain size modulates volcanic ash retention on crop foliage and potential yield loss**

Ligot Noa[1,*], Bogaert Patrick[1], Biass Sébastien[2], Lobet Guillaume[3,4], Delmelle Pierre[1]

[1]Environmental Sciences, Earth and Life Institute, UCLouvain, Louvain-la-Neuve, Belgium

[2]Department of Earth Sciences, University of Geneva, Geneva, Switzerland

6    [3]Agricultural Sciences, Earth and Life Institute, UCLouvain, Louvain-la-Neuve, Belgium

[4]Agrosphere Institute, IBG3, Forschungszentrum Jülich, Jülich, Germany

9    *corresponding author

Email: noa.ligot@uclouvain.be

Tel.: +32 (0)10 473 638

12    NL ORCID: 0000-0003-1416-3663



**Abstract**

15    Ash fall from volcanic eruptions endangers crop production and food security and jeopardises

agricultural livelihoods. As population in the vicinity of volcanoes continues to grow,

strategies to reduce volcanic risks to and impacts on crops are increasingly needed. This effort

18    involves the use of quantitative relationships for anticipating crop damage from ash exposure.

However, current limited models of crop vulnerability to ash rely solely on ash thickness (or

loading) and fail to reproduce the complex interplay of other volcanic and non-volcanic

21    factors that drive impact. Amongst these, ash retention on crop leaves affects photosynthesis

and is ultimately responsible for widespread damage to crops. In this context, we carried out

greenhouse experiments to assess how ash grain size, leaf pubescence and humidity

24    conditions at leaf surfaces influence the retention of ash (defined as the percentage of foliar

cover coated with ash) in tomato and chilli pepper plants, two crop types commonly grown in

volcanic regions. For a fixed ash mass load (~570 g m$^{-2}$), we found that ash retention

decreases exponentially with increasing grain size and is enhanced when leaves are pubescent

(such as in tomato) or their surfaces are wet. Assuming that leaf area index (*LAI*) diminishes

with ash retention in tomato and chilli pepper, we derived a new expression for predicting

potential crop yield loss after an ash fall event. A corollary result is that the measurement of

crop *LAI* in ash-affected areas may serve as a useful impact metric. Our study demonstrates

that quantitative insights into crop vulnerability can be gained rapidly from controlled

experiments, thereby providing a mean to improve models that can predict ash risks to crops

accurately. We advocate this approach to broaden our understanding of ash-plant interaction

and to validate the use of remote sensing methods for assessing crop damage and recovery at

various spatial and time scales after an eruption.



**Introduction**

The livelihood and food security of hundreds of millions of people living near and on volcanoes

intricately depend on agriculture (Small and Naumann, 2001; Brown et al., 2015). However,

farming activities in these regions is exposed to short-term, negative impacts of volcanic

eruptions, an issue amplified by the expanding population living under volcanic risk (Brown et

al., 2015; Freire et al., 2019). Widespread damage to agriculture during eruptive activity most

often arises from crop exposure to ash fall (e.g. Burket et al., 1980; De Guzman, 2005;

Tampubolon et al., 2018), causing adverse effects that range from temporary perturbations in

leaf physiology to irreversible mechanical damage (Eggler, 1948; Blong, 1984; Grishin et al.,

1996; Ayris and Delmelle, 2012). As a result, crop fields impacted by ash deposition produce

lower or poor-quality harvests that can translate into significant economic losses to farmers

(Neild et al., 1998; Wilson et al., 2007; Ligot et al., 2022).

In this context, the development of strategies that can support disaster risk reduction and

51 strengthen resilience for agrarian communities in volcanically active regions is critical,

especially in less-economically developed countries (FAO, 2021). Such measures require a

sound understanding of agriculture vulnerability to ash fall (UNDRO,1980; Jenkins et al., 2015;

Craig et al., 2021). Over the past 15 years, a dozen or so of post-eruption impact assessments

(post-*EIA*) have contributed to document the responses of farming systems exposed to ash (e.g.,

Wilson et al., 2007; Wilson et al., 2011; Magill et al., 2013; Blake et al., 2015; Craig et al.,

2016b; Craig et al., 2016a; Ligot et al., 2022). These field-based investigations have

underpinned the development of empirical relationships that link ash accumulation (also

referred to as ash mass load or deposit thickness) to an estimated level of production loss for

different agriculture types characterised by specific vulnerabilities (Wilson and Kaye, 2007;

Jenkins et al., 2014; Craig et al., 2021). In parallel, new methodologies harvesting the potential



of big Earth observation data and interpretable machine learning are being developed to

complement post-*EIA* studies (Biass et al., 2022).

Despite these recent efforts, current ash-loss of crop production relationships remain

overshadowed by uncertainties (Jenkins et al., 2015), which are rooted in three main sources.

Firstly, they lean on limited observational data, mostly acquired in post-*EIA* studies conducted

in temperate volcanic regions. Secondly, it is assumed that ground ash accumulation

(thickness or ash mass load) is the principal hazard intensity metric governing impact level on

crops. However, other volcanic (e.g. ash grain size, surface composition) and non-volcanic

factors (e.g. environmental conditions, plant traits, crop development stage) play a key role in

dictating impact and vulnerability (Jenkins et al., 2015; Ligot et al., 2022). Finally, current

approaches lack an impact metric that can be applied to anticipate crop damage from ash fall.

These limitations are hindering the development of accurate process-based risk assessment

models that can inform targeted strategies to reduce the risk of production loss in the case of a

volcanic explosive eruption.

Jenkins et al. (2022) estimated that an explosive eruption of *VEI* 4 (Volcanic Explosivity Index;

Newhall and Self, 1982) on the island of Java, Indonesia, has on average a 50% probability of

affecting ~700 km$^2$ of crops with ash. The surface area potentially affected by ash fallout is ~17

times larger for an eruption of *VEI* 5. Ash deposits thin exponentially from the source. Close to

the vent, ash fallout usually results in destructive impacts, where ash deposition exceeding

several cm in thickness may lead to smothering of the vegetation and direct mechanical

breakage of plant's parts (leaves, twigs, stem) (Ayris and Delmelle, 2012; Arnalds, 2013;

Jenkins et al., 2015; Craig et al., 2021). With increasing distance from the vent, impacts

gradually become disturbances. Thin ash blankets, able to affect several hundred to thousands

of km$^2$, retain the potential to cause serious crop yield loss without threatening plant integrity



(Magill et al., 2013; Ligot et al., 2022). In these areas, the capacity of ash fall to initiate damage

to crops hinges on the percentage of leaf surfaces covered by ash, here referred to as ash

retention. This relates to the shading effect exerted by solid particles deposited on leaves,

reducing light interception and decreasing photosynthetic activity (Thompson et al., 1984;

Hirano et al., 1995). Although ash grain size, leaf pubescence and ambient humidity have been

suspected to affect ash retention on foliage, accurately assessing widespread impacts on crops

from ash fall remains limited by the absence of a (i) systematic investigation of factors

controlling ash retention on foliage and (ii) quantitative impact metric reflecting crop

production loss.

Here, we adopt an experimental setup to investigate the influence of ash grain size, leaf

pubescence and humidity conditions at leaf surfaces on ash retention by crop foliage using

tomato and chilli pepper as model plants. By integrating the effect of both volcanic and non-

volcanic factors on ash retention, we formulate a novel conceptual model that uses *LAI* as the

99 impact metric for predicting crop yield loss when ash does not threaten plant integrity.

**Material and methods**

*Plant material and growing conditions*

Tomato (*Solanum lycopersicum* L.) and chilli pepper (*Capsicum annuum* L.) were chosen to

illustrate contrasting behaviours between plants of agronomical interest; they have a similar

stand in early growth period, but tomato has hairy leaves whereas chili pepper has glabrous

leaves. The seeds were sown in a sieved peat-based compost (pH 5-6.5) maintained at 24 °C.

Four weeks after sowing, the seedlings were transplanted in 1-litre plastic pots also filled with

peat-based compost. The average day and night temperatures in the greenhouse were 30 and

108 24 °C, respectively. Due to summer heats in Belgium, temperature during the day

occasionally rose above 35 °C. Combined with natural light, the use of *LED* lamps (120





µmoles m$^{-2}$ s$^{-1}$) provided a 16 h-photoperiod. Tomato and chilli pepper plants were watered

three times a week. They were exposed to ash six weeks after sowing, when tomato and chilli

pepper were at the seven- and eight-leaf stage, respectively.

*Simulated ash deposition*

We investigated the influence of ash grain size on the ability of tomato and chilli pepper

leaves to retain ash under dry and moist conditions. Six ash size ranges were tested, namely ≤

90, 90-125, 125-250, 250-500, 500-1000 and 1000-2000 µm. Each size range was tested in

combination with either dry or wet leaf surface conditions, i.e. a total of 24 treatments for

both crops. A treatment consisted of 15 replicates, corresponding to 360 measurements in

total. The ash material was obtained by crushing phonolite rocks (bulk composition: $SiO_2$ =

52.5, $Al_2O_3$ = 21.8, $K_2O$ = 9.6, $Na_2O$ = 7.8, $Fe_2O_3$ = 2.9, CaO = 1.5, $TiO_2$ = 0.3, MgO = 0.2

wt.%; density = 2.54 g cm$^{-3}$; Van Den Bogaard and Schmincke, 1984) obtained from a quarry

close to Laacher See volcano in Germany. The crushed phonolite was dry sieved for 10

minutes using an AS 200 Control Retsh vibrating sieve shaker with six sieves (90, 125, 250,

500, 1000, 2000 µm). The five size fractions coarser than 90 µm were wet sieved to remove

particles < 90 µm. The grain size distribution of the six ash size ranges was measured between

0.04 and 2000 µm by laser diffraction (Beckman Coulter LS13 320) (Fig.S1). The median

diameter was equal to 5, 98, 174, 401, 774 and 1465 µm for the ≤ 90, 90-125, 125-250, 250-

500, 500-1000 and 1000-2000 µm ash size ranges, respectively.

An ash load of ~570 g m$^{-2}$, corresponding to a deposit thickness of ~0.5 mm (i.e. considering

a deposit density of 1 g cm$^{-3}$), was applied uniformly to each plant using a homemade ash fall

simulator (Fig. S2). The device consists of a 135 cm-high *PVC* tube (of diameter 29.5 cm)

with three 1-mm opening meshes placed at 75, 110 and 120 cm from the tube base. Ash was

introduced evenly from the top of the tube through a 2 cm-mesh sieve. Wet conditions at leaf





surfaces were obtained by spreading ~1.5 g of water on each plant using a commercial manual

sprayer held one meter above the ground. In order to simulate the presence of water droplets

on plant leaves, we applied four sprays of water, one in each cardinal direction.

*Estimating the foliar cover from digital photos*

We took photos of each plant before and immediately after ash treatment. To minimise

uncontrolled variations in light colour and brightness, plants were photographed in a 1.6 x 1.2

x 2.2 m black box equipped with four led bulbs (6.5 W, cold white). We used a DX Nikon

camera with an AF-S DX NIKKOR 18-55mm f/3.5-5.6G VR II lens mounted on a 0.9 m-high

tripod. Sheets of paper were placed on the floor and plant pot to produce a uniform

background. A ribbon placed in a fixed position provided a reference scale.

We analysed the digital photos with ImageJ 1.52 (Schindelin et al., 2015) and wrote a macro

([https://github.com/NoaLigot/ImageJ-macro.git](https://github.com/NoaLigot/ImageJ-macro.git)) to estimate the foliar cover, which measures

the vertical projection of exposed leaf area. While digital photos are recorded as a raster of

147 red/green/blue (*RGB*) pixels, the values are not standardised and can vary depending on the

camera (Darge et al., 2019). The ImageJ macro transforms the *RGB* colour space into the

International Commission on Illumination (*CIE*) 1976 L*a*b* colour space (Mclaren, 1976),

which has linear measures of lightness (L*) and two colour dimensions (a* and b*). The a*

dimension represents a spectrum from green (negative) to magenta (positive) and the b*

dimension represents a spectrum from blue (negative) to yellow (positive). The a* attribute is

153 useful to identify green pixels and was used in the ImageJ macro to identify and select green

parts of leaves. Values of 1 and 0 are attributed to a green and non-green (background) pixel,

respectively. This allows delineation of the shape of the green leaf portion and calculation of

156 its surface area.

*Data treatment*



The percentage of foliar cover coated with ash was inferred for each plant by comparing the
foliar cover estimated from the image analysis, before and after ash application. Negative
percentage values (i.e. increase in green leaf surface after ash application) were obtained for
26 measurements, corresponding to treatments carried out with ash particles ≥ 250 µm. They
result from green leaf parts visible to the camera after leaves moved under the ash weight and
measurement errors linked to repositioning of the camera after ash application and
inaccuracies in the image analysis process. Negative values were all replaced with null values.
A Tukey *HSD* (Honest Significant Difference) test was applied to determine if means differ
between treatments. Tomato and chilli pepper plant measurements carried out under dry and
wet leaf surface conditions were processed separately, i.e. four sub-datasets were used in
order to compare the means separately for each combination of crops and moisture conditions.

**Results**

*Foliar cover coated with ash*

The percentage of foliar cover coated with ash ranged from 0 to 99%, with an average value
of 36 ± 33% (Table S1). The effect of ash grain size, humidity conditions at leaf surfaces and
leaf pubescence on the foliar cover coated with ash is illustrated in Fig. 1. In general, foliar
cover coated with ash increased with decreasing ash grain size. Grain size ≥ 500 µm covered
only 10% of the foliar cover, with coverage increasing up to ~90% for ash ≤ 90 µm. Wetting
of tomato and chilli pepper leaves prior to ash application had no effect on the retention of
fine ash (≤ 90 µm). Nevertheless, higher tomato and chilli pepper leaf surface coverages (+17
± 5% and +31 ± 10%) were inferred for intermediate ash grain sizes between 90 and 500 µm
(Table S1, S2). We also note that for the ash grain size ranges 125-250 and 250-500 µm in dry
conditions, coverage of tomato leaves by ash was on average greater by ~30 and 20%,
respectively, compared to chilli pepper leaves.





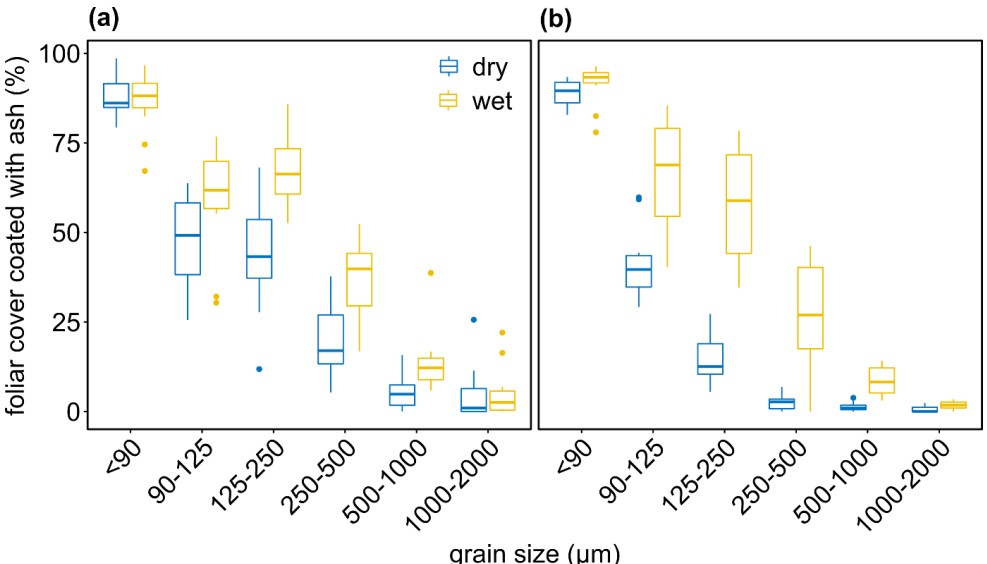

Figure 1: Percentage of foliar cover coated with ash for tomato (a) and chilli pepper (b) plants

as measured for the six grain size ranges tested in dry and wet conditions at leaf surfaces.

*Quantifying ash retention as a function of grain size*

Using the experimental results obtained for tomato and chili pepper (Fig. 1), we predicted the

percentage of foliar cover coated with ash as a function of grain size, when leaf surfaces are

dry or wet. Five convex models (i.e. exponential decay, power curve, rectangular hyperbola,

asymptotic curve and logarithmic curve) were fitted to the data points using the *aomisc* and

*nlme* packages in *R* (Onofri, 2020; Pinheiro and Bates, 2022) (Fig. S3). The median grain size

was used to represent the corresponding grain size range. A lack-of-fit sum of squares test

was applied to evaluate the relevance of each model. Since the five models have different

numbers of parameters, their test statistics (F*) could not be compared directly. Instead, the

models were assessed based on their *p*-values (Table S3). All the models have *p*-values > 5%,

with no evident lack-of-fit. The exponential decay model had the highest *p*-value for the four

sub-datasets (0.8, 1, 1, 1 for dry tomato, wet tomato, dry chilli pepper and wet chilli pepper,

respectively) and it was chosen for the predictions.





Quantile regressions using the exponential decay model indicate that for 500 µm ash particles,

there is a 50% chance to cover ~10 and ~27% of tomato foliar cover in dry and wet

conditions, respectively (Fig. 2). Similarly, for chilli pepper, foliar covers of <1 and 20% are

estimated in dry and wet conditions, respectively. By the same tenet, there is a 50%

probability that ash 63 µm in diameter covers up to ~67% (dry conditions) and ~77% (wet

conditions) of the foliar cover in tomato, and ~51% (dry conditions) and ~78% (wet

conditions) of the foliar cover in chilli pepper.

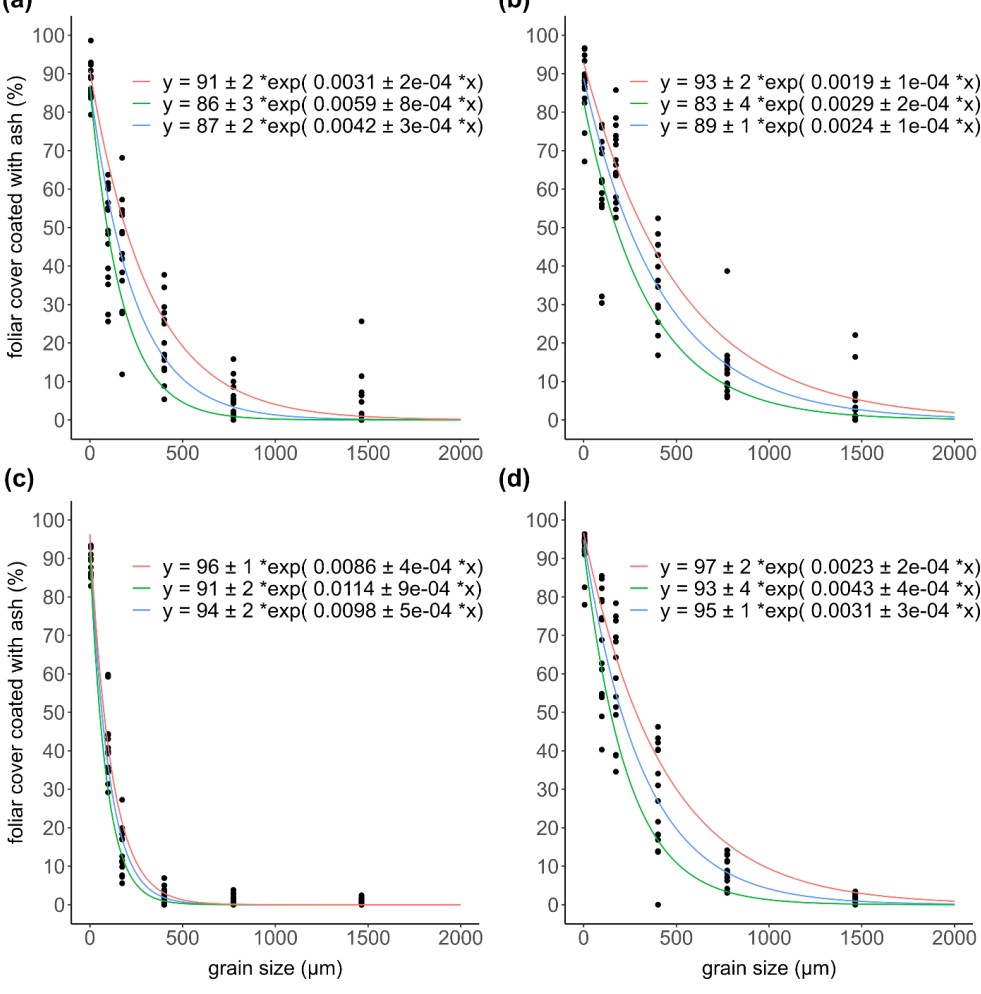





Figure 2: Quantile regression with the first quartile (green), median (blue) and third quartile

(red) for tomato plant in dry conditions at leaf surfaces (a), tomato plant in wet conditions at

leaf surfaces (b), chilli pepper plant in dry conditions at leaf surfaces (c) and chilli pepper

plant in wet conditions at leaf surfaces (d).

*Distribution of ash retention on the foliar cover*

In addition to controlling ash retention on leaves, grain size, conditions of humidity at leaf

surfaces and leaf pubescence affect the location of ash retention (Fig. S4). For tomato plants

in dry conditions, ash ≤ 90 µm tended to be lodged on the leaf surface wherever it had settled.

For glabrous chilli pepper leaves, leaf angle dictates if the ash particles remain on the leaf

surface after deposition or slide off and relocate elsewhere. Ash with intermediate grain sizes

between 90 and 500 µm behaved differently, depending on humidity conditions. For both

tomato and chilli pepper plants, the ash material was found mainly along the primary and

secondary veins of the horizontal upper leaves when they were dry. However, in wet

conditions, ash was more homogeneously distributed over the leaf surface. Coarser ash (≥ 500

µm) accumulated preferentially in the folds of growing leaves.





Figure 3: Images processed with ImageJ of tomato and chilli pepper plants after exposure to

~570 g m$^{-2}$ of ash varying in grain size ($\leq$ 90, 90-125, 125-250, 250-500, 500-1000, 1000-

2000 µm) and in dry and wet conditions at leaf surfaces. The part of the foliar cover depicted

in black corresponds to the green leaf surface area that was not covered by ash. The original

photos of the ash-covered plants are provided as supplementary material (Fig. S4).

**Discussion**

*Influence of grain size on ash retention*

The foliar cover coated with ash increases exponentially (from ~10 to 90%) when grain size

decreases from 500 to 90 µm, whether in dry or humid leaf conditions (Fig. 2). While the

exponential function inferred to describe this relationship was established for a single ash

mass load (~570 g m$^{-2}$), we anticipate a similar behaviour for lower or greater ash load values.

This result is in accordance with Miller (1967) and Johnson and Lovaas (1969) who found

that alfalfa, maize, bean, beet, cabbage, carrot, pea, pepper, potato, radish and squash exposed

to volcanic ash and quartz sand with grain sizes varying from < 44 to 350 µm was inversely





correlated with grain size. Witherspoon and Taylor (1970) reached a similar conclusion after

dusting various crops (i.e. squash, soybean, sorghum, peanut and clover) with quartz powders

differing in grain size (44-88 and 88-175 µm).

The fate of a solid particle falling from the atmosphere and hitting a leaf surface will depend

on how much of its initial kinetic energy is absorbed through tissue deformation (Vogel,

1989; Niklas, 1999; Benson, 2015). Ignoring aggregation processes, the coarser the particles,

the larger their terminal fall velocity and thus, kinetic energy (Dellino et al., 2005; Benson,

2015), simply reflecting that mass increases with grain size. If particles retain enough kinetic

energy after impact, they can bounce back and be ejected off the leaf or deposited elsewhere

(Gregory, 1961; Chamberlain, 1967; Starr, 1967; Chamberlain and Chadwick, 1972).

Otherwise, they will settle on the upper side of leaves, although they may be subsequently

displaced as new particles impinge the leaf surface. Based on the drag model for non-

spherical particles of Bagheri and Bonadonna (2016), we estimated the terminal fall velocity

of individual particles of 10, 100, 170, 410, 710 and 1470 µm, representing the median values

of the six ash size ranges used in our experiment. Terminal fall velocity increases with grain

size and is five times lower for particles of 100 µm diameter (assimilated to the fine ash

fraction) than for particles of 410 µm diameter (corresponding to coarse ash) (Table S4). This

result suggests that the kinetic energy of the finest ash particles is ~10,000 times smaller than

that of the coarsest material. The low kinetic energy of fine particles probably explains why

ash in the $\leq 90$ µm size fraction produces a greater foliar cover compared to ash $\geq 500$ µm

(Fig. 2). In contrast, coarse ash particles with higher kinetic energy will tend to lodge on less

elastic leaf structures, such as primary and secondary veins and folds (Fig. 3).

*Influence of leaf pubescence on ash retention*



On average, ash particles in the intermediate size range 125-500 µm cover ~25% more foliar cover in tomato than in chilli pepper (Fig. 2, Table S1). This is attributed primarily to the presence of leaf hairs in tomato. Sæbø et al. (2012) and Ram et al. (2012) demonstrated that dust accumulation on the foliage of various trees and shrubs is proportional to leaf hair density. Leaf hairs enhance dust collection area and capacity to absorb the falling particles' kinetic energy. In addition, leaf pubescence may prevent particles from sliding off the leaf surface. By increasing friction on particles, leaf hairs counteract the gravity force generated by mass loading on the leaf surface which pulls a leaf downward (Smith and Staskawicz, 1977). In our experiments, ash ≤ 90 µm adhered to the tip of pubescent leaves with a steep inclination angle in tomato plants, whereas it barely encroached on the glabrous surface of chilli pepper leaves (Fig. 3). Previous field observations of ash-impacted crops also highlight a stronger adherence of ash on pubescent leaves (such as barley, corn, tobacco, tomato and apple tree) and hairy fruits (such as peach, apricot, kiwi-fruits, strawberry and raspberry) (Miller, 1967; Cook et al., 1981; Wilson et al., 2007; Sword-Daniels et al., 2011; Ligot et al., 2022). Witherspoon and Taylor (1970) concluded that the pubescent leaves of squash and soybean favour a uniform retention of quartz particles (88-175 µm). In contrast, the glabrous leaves of rose plants exposed to the 1963 eruption of Irazu volcano, Costa Rica, collected little ash material (Miller, 1967).

*Influence of humidity conditions at leaf surfaces on ash retention*

Wetting of leaves prior to application of ash with an intermediate grain size of 90-500 µm increased the foliar cover coated with ash of tomato and chilli pepper by $17 \pm 5\%$ and $31 \pm 10\%$, respectively (Fig. 2, Table S2). We also noted that the ash deposit that formed on pre-wetted leaves appeared more homogeneous compared to that observed when the leaf surface was dry (Fig. 3). Similarly, Miller (1967) reported during the 1963 eruption of Irazu that wet





leaf surfaces facilitated retention of ash < 300 µm and formation of a homogeneous deposit.
Enhanced ash retention on wet leaves likely relates to the surface tension generated by water

molecules present on the leaf surface (Tabor, 1977; Israelachvili, 2011).

*Modelling potential yield loss in tomato and chilli pepper plants exposed to ash*

Our experimental results show that fine ash can readily cover the upper side of leaves (Fig. 2).

Assuming an ash material comprised of spherical particles 90 µm of diameter and with a

density of 2.54 g cm$^{-3}$ (i.e. the density of phonolite), we calculated that a mass load as low as

~8.6 g m$^{-2}$ can form a monolayer deposit on a leaf surface. While this estimate represents an

oversimplified situation, it is more than fifty times less the ash load (~570 g m$^{-2}$) used in our

experiment. Since fine particles are ubiquitous– albeit in various proportions – in ash fallout

(Rust and Cashman, 2011; Costa et al., 2016), an ash coating on leaf surfaces is likely to form

in areas affected by explosive eruptions. Importantly, the presence of solid particles on foliage

exerts a shading effect, which reduces light interception (*LI*, dimensionless) by leaves

(Thompson et al., 1984; Hirano et al., 1990). For example, Hirano et al. (1991) measured a

~20% decrease in *LI* after treating mandarin tree leaves with only 4 g m$^{-2}$ of road dust (0.1-

100 µm). Similarly, deposition of 10 g m$^{-2}$ of ash (0-100 µm) on cucumber plants led to a

~20% reduction in *LI* (Hirano et al., 1992).

Recalling that *LI* drives net photosynthesis rate and thereby, total biomass production

(Wilson, 1967; Biscoe et al., 1977; Monteith, 1977; Weraduwage et al., 2015), we contend

that even a thin ash deposit on crop leaves can drive yield loss. Thus, the interference of ash

with *LI* provides an indirect mean to predict the potential crop production loss for ash mass

loads below the threshold (cm-thick deposit) of direct mechanical damage to plants. Although

we did not measure *LI* in our experiment, this parameter can be inferred using the following

expression (Monteith, 1969):





$$LI = (1 - e^{-k \times LAI}) \tag{1}$$

where $k$ is the light interception coefficient (dimensionless). The temporal evolution of $LAI$

during plant growth has been documented for tomato and chilli pepper in various studies (e.g.

Campillo et al., 2010; Monte et al., 2013; Al Mamun Hossain et al., 2017; Mendoza Perez et

al., 2017) and this information allows the estimate of $LI$ using Eq.(1) (see Supplementary

material).

The daily biomass accumulation by crop canopy ($CBIO_c$, g m$^{-2}$ day$^{-1}$) depends on $LI$

according to (Monteith, 1972; Hatfield, 2014):

$$CBIO_c = Q \times LI \times RUE \tag{2}$$

where $Q$ is the incident radiation (MJ m$^{-2}$ day$^{-1}$) and $RUE$ (g MJ$^{-1}$) the radiation use

efficiency. Representative values for $Q$ in Belgium (warm temperate humid climate) and $RUE$

are available from the scientific literature (Table S5). The crop harvested biomass ($CBIO_h$, g

m$^{-2}$ day$^{-1}$) is calculated as the sum of the $CBIO_c$ in the time period considered (i.e. number of

days elapsed between transplanting and harvest) multiplied with the harvest index ($HI$,

dimensionless) (Kemanian et al., 2007; Hay, 2008):

$$CBIO_h = \sum_{sowing}^{harvest} CBIO_c \times HI \tag{3}$$

Figure 4 depicts the concepts underpinning Eqs. (1), (2) and (3).

We hypothesised that $LAI$ reduction in crop plants exposed to ash is directly proportional to

the percentage of foliar cover coated with ash deposits (Fig. 2), presupposing that ash-affected

leaves lose their ability to perform photosynthesis efficiently. Based on this, and using Eqs.

(1), (2) and (3), potential crop yield loss ($CYL_\%$, %) can be deduced by comparing the

harvested biomass in the absence ($CBIO_h^{no\ ash}$) and presence ($CBIO_h^{ash}$) of ash:



$$CYL_\% = 100 \times \frac{CBIO_h^{no\ ash} - CBIO_h^{ash}}{CBIO_h^{no\ ash}} \tag{4}$$

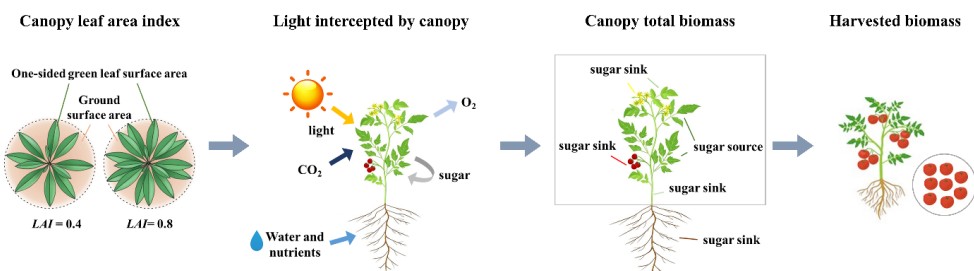

Figure 4: Cartoon conceptualising the relationships between canopy leaf area index (*LAI*), light interception by canopy, canopy total biomass and harvested biomass.

To illustrate our approach, we estimated $CYL_\%$ for tomato and chilli pepper plants exposed to

~0.5 mm (or 500 g m$^{-2}$) of ash. We tested different ash size distributions and evaluated the

influence of humidity conditions at leaf surfaces on ash retention. Two scenarios of plant

exposure to ash fall were considered: one in which 25% of the plant growth period is

completed (i.e. 32 days after transplanting for tomato and 57 days after transplanting for chilli

pepper), and one in which 75% is achieved (i.e. 97 days after transplanting for tomato and

172 days after transplanting for chilli pepper). The daily *LAI* evolution of tomato and chilli

pepper plants during growth was computed in *R* using published data (Fig. S5).

In our model, the entire plant canopy received the same amount of ash, although some leaves

may be less exposed due to their position on the stem. We also considered that ash deposition

on leaves neither halt plant growth nor production of new leaves and therefore, *LAI* can

recover after the ash fall event. The calculated temporal evolution of the *LAI* of tomato plant

that has completed 25% of its growth period when it receives ash (90-125 µm in diameter,

mass load of ~570 g m$^{-2}$) in dry conditions is illustrated in Fig. 5a. A similar temporal

evolution of *LAI* is obtained for chilli pepper (Fig. S5).





The presence of ash on plant canopy may lead to premature leaf senescence (as reported by

Miller, 1967; Neild et al., 1998; Wilson et al., 2007; Ligot et al., 2022), impacting $CBIO_h$ (Eq.

3). To account for this effect, we subtracted the ash-coated leaf biomass from the total canopy

biomass, the latter being comprised of the leaves and stem. For tomato and chilli pepper

plants, leaf biomass represents ~60% of canopy biomass (Kleinhenz et al., 2006; Elia and

Conversa, 2012; Poorter et al., 2015). The leaf biomass fraction affected by ash can be

inferred from Fig. 1. Resolving Eqs. (1) and (2), the temporal evolution of $CBIO_c$ for tomato

or chilli pepper subjected to ash can be predicted. Fig. 5b illustrates this for tomato plant

exposed in dry conditions to ash deposition (90-125 µm in diameter; mass load of ~570 g m$^{-2}$)

32 days after transplanting (i.e. at 25% of growth period). Since the leaf-to-canopy biomass

ratio and percentage of leaf biomass covered by ash which dies are set equal for both crops, a

similar trend is inferred for chilli pepper (Table S5)





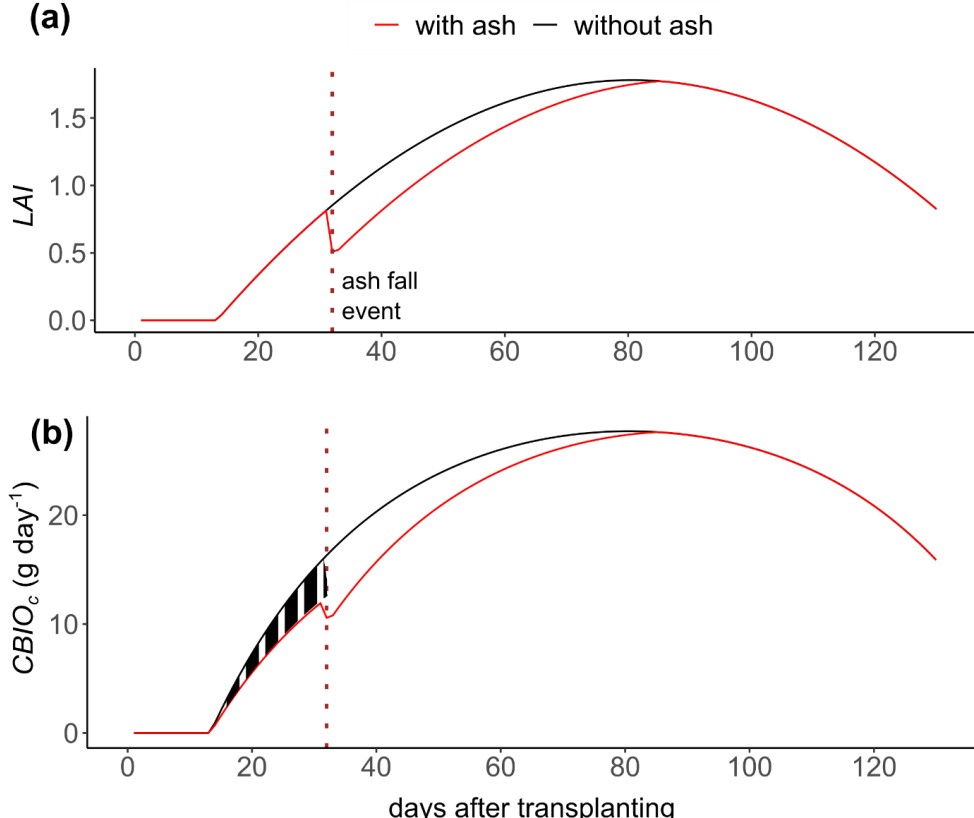

Figure 5: Temporal evolution of the leaf area index (*LAI*) (a) and daily biomass accumulation

(*CBIO$_c$*) (b) of tomato plant exposed to ~570 g m$^{-2}$ of ash (size range: 90-125 µm) 32 days

after transplanting (i.e. at 25% of the growth period) in dry leaf surface conditions. The

hatched area represents the leaf biomass produced by the plant before the ash fall event and

which will undergo premature senescence after it. The ash covered leaf biomass is inferred

from the leaf-to-canopy biomass ratio (i.e. 60%) and the percentage of leaf biomass covered

by ash (i.e. 48% for tomato in dry leaf surface conditions).

As detailed above, ash impact on *CBIO$_h$* is modulated by different factors, including the *LAI*

fraction that becomes photosynthetically inactive due to the presence of ash coatings on

leaves (i), number of days elapsed between ash deposition and emergence of new leaves (ii),





leaf-to-canopy biomass ratio (iii), and percentage of leaf biomass covered by ash and which

eventually dies (iv). Our model calculations revealed that crop growth period determines the

relative importance of each of these factors in determining $CYL_\%$. For example, if 90 µm ash

affects tomato and chilli pepper plants in dry conditions at 25% of their growth period, $CYL_\%$

is most sensitive to (i) and (ii), whereas for older plants that have completed 75% of their

growth, (iii) and (iv) are the main factors driving $CYL_\%$ (see Supplementary material).

In order to assess the error on $CYL_\%$ estimates, we applied a stochastic approach with 10,000

simulation runs using a random value for each of the four factors (as listed above) that can

influence the final model output. We posited that the values taken by factors (iii) and (iv)

follow a gaussian distribution (Table S5), whereas variable (i) and (ii), which are always in

the range 0-1 and positive, respectively, are described by a truncated gaussian distribution.

Fig. 6 shows the uncertainties on $CYL_\%$ as computed by fitting the first and third quartiles

around the median $CYL_\%$ value for tomato exposed to ash of different grain sizes, either in dry

or wet leaf conditions. Calculations were repeated for plants that receive ash when at 25 and

75% of their growth period. For tomato, $CYL_\%$ increases with decreasing ash grain size (Fig.

6). Tomato plants at 25% of their growth may experience a 2-17% decrease in yield

depending on grain size and humidity conditions at leaf surfaces. A significantly higher $CYL_\%$

(0-42%) is anticipated when ash affects plants at 75% of their growth. A similar pattern

emerges for chilli pepper where $CYL_\%$ varies between 1-17 and 0-46% when considering that

the plant receives ash when at 25 and 75% of its growth period, respectively (Fig. S6). For

intermediate ash grain sizes between 125 and 500 µm, the $CYL_\%$ is 5, 3, 8 and 4% greater for

tomato compared to chilli pepper when exposure to ash occurs at 25% of the growth in dry

conditions, 25% of the growth in wet conditions, 75% of the growth in dry conditions and

75% of the growth in wet conditions, respectively.



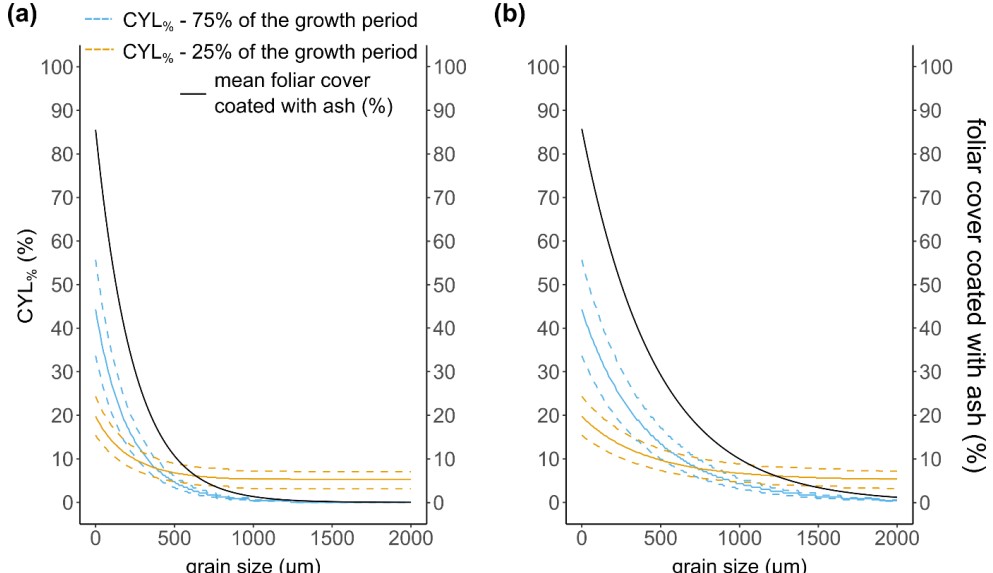

Figure 6: Potential crop yield loss (*CYL%*, first quartile, median and third quartile) estimated
for tomato plant as a function of ash grain size in dry (a) and wet (b) conditions at leaf
surfaces.

*Towards using LAI as an impact metric for predicting potential yield loss in ash-affected*
*crops*

While deployment of field-based post-*EIA* will continue to enrich our understanding of ash-
loss of production relationships, progress is contingent on eruption occurrence, site
accessibility, limited field time, variations in environmental conditions and incomplete ranges
of ash characteristics such as thickness and grain size (Jenkins et al., 2015). Here, we have
shown how empirical data from experimental testing can be transformed into quantitative
insights for predicting potential yield loss in tomato in chilli pepper exposed to ash. Our
model identifies that reduction in *LAI* following ash deposition ultimately drives reduction in
production. Changes in *LAI* in ash-affected crops is interpreted in terms of a shading effect
and *LI* reduction; ash retention on leaves being influenced by grain size, plant traits and
environmental conditions (Fig. 1). As detailed in Eqs. (1), (2) and (3), crop yield depends on




*LAI* and therefore, the latter is regarded as an integrative impact metric. From this, we propose

that *LAI* measurements in crop plants subjected to ash fall offer a new mean for analysing

crop vulnerability and forecasting potential yield loss for ash mass loads below the threshold

(cm-thick deposit) of direct mechanical damage to plants. The rapidly increasing ability to

monitor crop characteristics, including type, *LAI* and biomass, using optical and radar earth

observation data (Hosseini et al., 2015; Fang et al., 2019; Rosso et al., 2022) provides an

unprecedented opportunity to collect a spatially- and time-resolved information that can

support the development of more realistic and more complete ash-loss of crop production

relationships.

In order to unlock the full potential of *LAI* estimates for investigating the vulnerability of

crops to ash events, more knowledge on how ash coatings on leaves interfere with *LI* is

required. In our model of potential yield loss in tomato and chilli pepper (Fig. 6, S5), we

equated *LAI* reduction with the foliar cover percentage covered by ash. In essence, this means

that an ash deposit on leaves renders light interception inoperative. This may not always be

the case because *LI* by a crop canopy is determined not only by the *LAI* of the species, but

also by the light absorption characteristics of the leaves (Liang et al., 2012), here modified by

the ash coating. Further laboratory investigations can generate the empirical observations

needed to better constrain the changes in *LI* in relation to the characteristics (thickness/mass

load, grain size, albedo) of the ash material deposited onto the leaf surface.

The evolution of *LAI* following an ash deposition event (Fig. 5a) was modelled by assuming

that ash-affected plants will grow new leaves after a set period of time. Our analysis showed

that $CYL_\%$ is sensitive to this parameter, therefore requiring adjustment depending on crop

type (Klepper et al., 1982). We also note that many crops (including major ones such as

wheat; Hay and Porter 2006) have a determinate growth habit and as such, may not be able to





sprout new leaves if they receive ash late in their development cycle. Thus, the effect of ash

fall on crop *LAI* hinges both on plant growth characteristics and timing of the volcanic

eruption.

We considered in our model that an ash deposit induces premature leaf senescence, in

agreement with field observations (Miller, 1967; Neild et al., 1998; Wilson et al., 2007; Ligot

et al., 2022). While this process probably relates to leaf chlorosis due to *LI* reduction

(Bilderback 1897; Mack, 1981; Ligot et al., 2022), its temporality and precise mechanism

remain unclear. New experimental investigations with various crop plants will help to better

constrain the proportion of leaf biomass affected by ash which will be subjected to premature

senescence.

We have highlighted that grain size, leaf pubescence and humidity conditions at leaf surfaces

control ash retention, which in turn drives *LAI* reduction. Other factors may influence ash

retention. For example, leaf microstructural features such as stomatal density and presence of

a waxy epicuticle have been shown to influence retention of non-volcanic dust particles

(Sæbø et al., 2012; Zhang et al., 2017). In addition, in the natural environment, wind- and

450 rain-driven erosion processes can remove ash deposited on foliage. Conversely, light rain may

induce crusting of ash, prolonging its residence time on leaves (Miller, 1966; Ayris and

Delmelle, 2012; Le Pennec et al., 2012; Ligot et al., 2022). The significance of these

453 environmental variables in controlling ash retention time by leaves has never been assessed

quantitatively, calling for further field and experimental investigations.

Finally, our approach for modelling production loss in tomato and chilli pepper exposed to

456 ash assumes that light interception is the main variable governing plant growth. While this is

true in our study where water and nutrient supply were never limited, more stringent

conditions may be encountered in crop fields subjected to ash fall. For example, an ash layer



on the ground may alter water and gas movements into and through the soil and surface runoff

(Ayris and Delmelle, 2012; Neslon, 2013; Tarasenko, 2018), in turn impacting the soil water

balance. A better comprehension of the side effects of ash depositions on the soil plant-system

is needed in order to identify the primary mechanisms driving the short- and long-term

consequences for crop production.

**Conclusions**

Our study highlights the usefulness of conducting experimental measurements to supplement

observations obtained from post-*EIA*. It provides a new perspective into the volcanic and non-

volcanic factors that control ash impact on crops. The experimental results obtained for

tomato and chilli pepper plants demonstrate that ash retention on leaf surfaces increases with

decreasing grain size and is enhanced when leaves are pubescent and wet. We also showed

that, for a given ash mass load, the percentage of leaf surfaces covered by ash is an

471 exponential decay function of grain size, the parameters of this function being influenced by

leaf pubescence and humidity conditions at leaf surfaces. Thus, we conclude that the

proportion of fine material in ash fallout is an important hazard metric for assessing risk to

474 crops. The corollary to this finding is that relying on ash thickness (or mass load) alone to

anticipate crop damage from ash is inaccurate and possibly misleading.

Using the empirical relationship linking ash retention to ash grain size and equating ash

retention with *LAI* reduction, we have developed a novel model framework to predict *CYL%*.

This approach identifies *LAI* as a promising impact metric that can be quantified for assessing

crop production following an ash fall event. *LAI* is commonly retrieved *via* remote sensing

measurements. The rapid deployment of new satellites allows data collection at increasingly

high spatial and temporal resolution (for example, the European Space Agency's Sentinel-2

mission), paving the way for estimating *LAI* at the crop field scale. Additionally, the



technology gives access to FPAR, i.e. the fraction of the solar radiation absorbed by live

leaves for the photosynthesis activity, which should also record a reduction in light

interception for leaves covered with ash. We anticipate that tapping into satellite-derived

measurements will considerably improve our quantitative understanding of crop vulnerability

to ash fallout. However, for exploiting their full potential, field- and laboratory-based

validations are required, including experiments aimed at constraining *LI/LAI* reduction in

relation to ash retention and characteristics. Acquiring this knowledge will significantly

enhance our capacity to accurately estimate ash risks to crops and thus, will help informing

the development of efficient risk mitigation strategies in agricultural regions exposed to

492 volcanic eruptions.

**Code availability**

The Image J macro to analyse the plant photos and estimate the foliar cover coated with ash

and the R script to compute the daily tomato and chilli pepper *LAI*, *LI*, $CBIO_c$ and $CYL_\%$ are

available on GitHub (https://github.com/NoaLigot/ImageJ-macro.git and

https://github.com/NoaLigot/R-scipt-LAI-LI-biomass-yield-loss/blob/main/script,

respectively).

**Data availability**

All raw data can be provided by the corresponding authors upon request.

**Author contribution**

NL, PD and GL conceptualized the experiments and NL carried them out. PP advised on the

statistical analysis and modelling approach. NL analysed the data, wrote the R script and ran

the simulations with the help of SB. NL and PD wrote the original draft with contributions

from all co-authors. PD secured funding for this research and provided the resources.



**Competing interests**

The authors declare that they have no conflict of interest.

**Acknowledgements**

N.L.'s doctoral research is supported by the FSR-FNRS (Fonds National de la Recherche

Scientifique 1.E077.19). N.L. is grateful to VOCATIO for a Fonds Ernest Solvay award that

contributed to support this study. This work was partly funded by a UCLouvain FSR-ARC,

"Talos" research grant (20/25-106). N.L. and P.D. are indebted to Marc Migon (SEFY, Earth

and Life Institute) for technical assistance, Xavier Draye (Earth and Life Institute) for lending

the camera equipment and Karen Fontijn (Department of Geosciences, Environment and

Society, Université Libre de Bruxelles) for access to ash sieving facility.



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
