# Peer review of "Grain size modulates volcanic ash retention on crop foliage and potential yield loss"

_EGUsphere, 2022_

## Referee Comment (RC1)

**Review of 'Grain size modulates volcani ash retention on crop foliage and potential yield loss' by Ligot et al.**

I read with great interest the manuscript submitted by Ligot et al. reporting on the results of lab experiments in which tomato and chili pepper plants are exposed to ash fallout of various grainsize, and the effect on the coverage of leaves is quantified. The manuscript proposes a relationship linking ash grain size and foliage cover, and further model the impact on potential yield loss. I am not a bio-engineer so I am not able to comment on the relevance and potential shortcoming of this later modelization, but I recognize the scientific interest in translating a hazard into potential consequences.

The manuscript is very well written and structured, it is rather concise and illustrated by well-designed figures. Reference is being made to many supplementary material: whereas for most it is justified to propose this material as supplement, some figure might better fit into the main manuscript.

The proposed methodology and the obtained results are quite original, at least for the volcanology discipline, and could have relevance for risk assessment; I therefore advise that this manuscript should be accepted for publication in NHESS after minor revisions are implemented in the manuscript, based on the comments made hereunder.

LITERATURE

In the introduction, information from post-eruption field assessment are shortly reviewed, as well as some papers focusing on the effect of dust/ash on plant more specifically (line 74-90). However, in the discussion, the authors point out to the existence of previous field and experimental work more specifically related to their research (by Miller et al., Johnson et Lovaas, Hirano et al. ). It is needed to shortly mention these previous research in the introduction to justify that 'ash grain size, leaf pubescence and ambient humidity have been suspected to affect ash retention on foliage' (lines 91-92);

EXPERMENTAL PROCEDURE

The methodology section describe in quite some details the novel experimental protocol implemented in this research. However, some elements could be further clarified. Additionally, a figure (or photo) representing the entire setup, including the device to spread the ash and the imaging system, would help the reader to understand the protocol.

- Line 111: specify the typical height of plants and typical surface areas of leaves at the time of experiments;
- Line 119: crushing phonolite rocks: did you check the morphology of the produced particles using SEM after crushing and compared to actual ash? Although I understand that crushing might be the best way to generate large ash fraction, the morphology of particles might influence their interaction with leaves.
- Ash loading of 570 g m$^{-2}$ (line 129): why did you select such loading that could be considered very low. Could it be expected that at larger loading, larger surface of leaves be covered until the grainsize does not matter anymore (see main comment below). Further discussion on the potential impact of ash loading on the observed relationship would be useful.
- Line 133: "through a 2 cm-mesh sieve" – is this correct? 2 cm seems extremely coarse relative to the ash used and would not help to distribute the ash evenly across the device.
- Lines 135 and following: the protocols should be more specific for what concern the timing and location of different actions. What was the duration between the spraying of water on

the leaves (for wet condition), the spreading of the ash and the acquisition of the pictures? Did these action follow each other within minutes or were there hours/days in between? Was spreading of ash conducted at the same location of the acquisition of the picture or was the plant displaced?

- Line 145: precise here that image is acquire before and after ash fallout (mentioned later on, but needed here for better understanding).
- Ash retention: was there any way to quantify the proportion of ash that was retained on the leave versus the ash that reached the ground? Were the plant weighted before and after the ash fallout?
- Lines 160-165: issue of leave bending. Authors report that some of their measurement returned higher 'green leaf surface' after ash exposure than before, claiming that this is due t movement of leaves and camera during image acquisition. As these issues probably affected all their measurement, the accuracy of the documented covered leave surface could be derived by considering the noise observed for experiments were the retention is close to zero. Additionally the issue of leave bending should receive further attention in the description of results: did significant bending or change in orientation of leaves were observed? For which grain size? Beyond the impact on the imaging procedure, the bending would also directly influence the potential of retention of ash? This is mentioned on line 266 ('which pulls a leaf downward') but no comment is made on whether this process was observed during experiments;

OTHER FACTORS

- **Ash loading**: authors decide to work with a single ash loading for all experiments. They properly argue that they select an ash loading that is below the threshold for physical damage for the plant (is such threshold well defined? Is it plant specific?). Assumption is made that the relationship between grainsize and foliage cover found for this ash loading would be valid also for other loading (or at least the type of relationship – lines 231-32). Would the retention of ash not relatively increase with increasing ash loading? Until a point were all the leave surface are covered irrespective of grainsize?
  Could it be assumed that once a first layer of ash is retained on the vegetation, the effect of grainsize on accumulation would not be valid, the ash particles creating their own roughness at the surface? Further discussion on the ash loading for which the observed role of grainsize might be valid should be further discussed. Similarly the reader should be reminded that the yield loss mentioned are only valid for the ash loading used in the experiment and that ash loading will most probably be a significant parameter in controlling foliage cover.
- **Residence time of ash**: very limited attention is giving to the time component; Authors consider the timing of the ash fallout relative to the growth of the plant, but not the duration of the ash retention on the leave (assuming early senescence of ash covered leaves). As the duration of residence not been considered in previous study? For how long does the ash need to cover the leave to cause decay? In intro (line 87-88) and discussion (line 449-450) this issue of duration should be shortly mentioned (in relation to wind/rain 'erosion')
- **Physical integrity**: authors systematically mention that they consider impact of ash on foliage for loading below the loading required to affect 'plant integrity' (line 99). However this threshold is not clearly defined (line 304: 'cm-thick'). I guess this threshold will be specific for each plant and development stage of a plant. This could be further clarified in discussion,

IMPLICATIONS

In both the introduction (lines74-75) and conclusion (line 491), authors claim that understanding and quantifying the retention of ash on crop foliage represent an essential step in mitigating the impact of eruption on agriculture. I agree that the presented results will contribute to better assess quantitatively the potential impact of ash fallout on crops (reduced yield), however it is unclear to me what the author consider as potential mitigation measures that could be derived from these results. The mitigation actions should be specified or the focus should be on the impact assessment.

SMALL EDITS

- Abstract is well written but could be shortened both in the problem statement and the results implication
- Line 41: 'farming activities ARE exposed'
- Line 48: 'economic loss' – in country with subsistence farming the issue of food shortage would also have to be considered.
- Line 76-79: which ash thickness/loading is considered to calculate these areas of crop affected?
- Figure 1: specify the number of experiments represented by each boxplot (is it 15?). Explanation of how to read the box plot (median, 25-75th quantile) should be added to caption.
- Figure 3: add scale bar or specify the area imaged in the caption.
- Line 285: figure 1 highlight that surface wetness has more influence on retention for chili pepper than tomato plan. This observation should be discussed here: I guess that leave pubescence and wetness act in a similarly way, so that wetness induces lower additional retention with tomato plants
- Line 320: explain what is the 'harvest index'
- Line 335-340: explain here how the impact of ash on the plant growth is simulated through leave senescence followed by new leave growth.
- Figure 6: provide also the results for chili pepper in the main text, these are important results.

---

## Referee Comment (RC2)

**Review of 'Grain size modulates volcanic ash retention on crop foliage and potential yield loss' Ligot, Bogaert, Biass, Guillaume, & Delmelle**

This manuscript presents both an experimental methodology of determining the change in leaf area index (LAI) due to volcanic ashfall, as well as a quantification of the relationship between deposit grain size and foliar coverage for tomato and chilli pepper plants. This is of significant interest as we work towards improving our ability to quantify agricultural losses based on characteristics of the ashfall deposit that productive land is exposed to. As noted in the manuscript, these assessments rely primarily on ashfall thickness (and/or loading) despite increasing recognition that other ashfall properties also have significant influence on the impacts received.

The manuscript writing, structure, and figures are all of a high standard suitable for publication. There is a large amount of supplementary material referred to throughout the manuscript, some of this may be better displayed as in-text figures/tables.

However, it would benefit from some clarification on aspects of the methodology - particularly the timing and location of steps, and further justification of why certain actions were chosen. Additionally, the broader applications of this approach at a larger scale using remote sensing, and with varying ash deposit thicknesses and compositions is not clear. The use of this approach to inform mitigation strategies is also not explained, and it is not obvious how this work would add to or enhance current approaches to mitigation. Another limitation that needs acknowledging is the use of only one very thin ash thickness and the potential influence of leaf/plant orientation to prevailing winds and the volcano.

However, the methodology presented is novel and could be expanded upon to enhance our agricultural impact assessment capabilities. Therefore, I believe this manuscript should be accepted for publication in NHESS once minor to moderate revisions are undertaken.

**Abstract**
Line 15 and throughout: 'ash fall' is commonly one word 'ashfall'
Line 33: 'mean' should be 'means'

**Introduction**
Line 40: 'is' should be 'are'
Line 40: Define 'short-term'
Line 44: True in areas where cropping farming dominates (e.g., Indonesia) but not in other countries where pastoral farming of livestock dominates
Line 48: Also food security issues in areas where farming is subsistence
Line 63: Expand/give examples
Line 67: I think its fair to say that tropical and semi-arid areas are increasingly being considered
Line 72: Is there really no impact metric? Isn't thickness/loading used in this way currently? It's not perfect but it is still indicative of likely crop damage to some extent – as you use it to eliminate the possibility of structural damage later in the manuscript
Line 81: 'cm' to 'centimetres'
Line 84: Insert 'less severe' before 'disturbances'

Line 84: Change 'blankets' to 'deposits'

Line 85: Change 'km$^2$' to 'square kilometres'

Line 85: Insert 'structural' before 'integrity'

Line 88: Is this always true? Reference? Wouldn't the depth of cover or leachable chemistry of the deposit sometimes be the mechanism of loss?

Lines 90-91: Evidence to support this point/reference needed

**Materials and methods**

Line 103: Rationale for choosing these two plant types needed

Line 108: Clarify that the experiments took place in Belgium

Line 112: Limitation that all plants the same age needs to be acknowledged. What height and leaf sizes?

Lines 119-128: What was the morphology of the particles in relation to natural ash deposits? No surface chemistry on synthetic ash material – does/does not influence ash retention and adherence?

Line 129: Only one very thin ash deposit thickness (~0.5 mm) tested. Ash thickness effect on retention not considered.

Line 131: Where is Fig. S1 in-text reference

Line 136: Did you immediately dose with ash after spraying?

Line 138: Were the plants moved between ashfall and photography? Or was the ash applied in the photography box?

Lines 138-156: How would this method scale up for use in a real-world situation? Needs to be included in discussion

Lines 161-164: Wouldn't this limitation apply to all measurements taken in this study? Any idea of the magnitude of this impact on the results? How is this accounted for?

**Results**

Line 173: How is leaf pubescence included in Fig. 1?

Line 176: Add 'significant' before 'effect'

Lines 183-184: Explain the points and boxplots in the caption

Line 201: Change 'that ash 63 µm in diameter' to 'that ash with a median of 63 µm in diameter'

Figure 2: Did leaf pubescence influence these curves? Was there enough data to quantify this?

Lines 219-220: Link this to the function of these parts of the leaves in the discussion

Figure 3: Could before photos be included too? The figure needs a scale

**Discussion**

Lines 233-236: Was this experimental or field data?

Line 243: Only true if considering a homogenous ash composition

Line 285: Is spraying the leaves with water an accurate representation of common humid environment?

Line 289: How does the density of the phonolite used compare to the density of other ash deposits?

Line 300: Change 'Recalling' to 'Considering'

Line 304 and elsewhere: Define the 'cm-thick deposit' threshold specifically

Line 317: How does the Q value for Belgium compare to Q values for more commonly volcanically active countries?

Line 320: Define 'harvest index'

Line 342: Evidence/reference that 'ash deposition on leaves neither halt plant growth nor production of new leaves…'

Lines 357-359: Why are these equal?

Figure 5: Show the same graphs for chilli pepper plants in this figure

Lines 379-381: Why were these distributions selected?

Line 405: Change 'in chilli pepper exposed' to 'and chilli pepper crops exposed'

Line 412: Change 'mean' to 'method'

Lines 414-419: More information on this is needed to demonstrate how the approach can be scaled up from a greenhouse set-up, please

**Conclusions**

Lines 489-492: It is unclear how this method and its results would add anything to existing mitigation efforts. Needs further explanation on the practical ways that this data could assist in an event

---

## Author Response (AR1)

**Reviewer 1's comments**

LITERATURE

**R1.1**

In the introduction, information from post-eruption field assessment are shortly reviewed, as well as some papers focusing on the effect of dust/ash on plant more specifically (line 74-90). However, in the discussion, the authors point out to the existence of previous field and experimental work more specifically related to their research (by Miller et al., Johnson et Lovaas, Hirano et al. ). It is needed to shortly mention these previous research in the introduction to justify that 'ash grain size, leaf pubescence and ambient humidity have been suspected to affect ash retention on foliage' (lines 91-92)

The following references have been inserted:

Line 97: Thompson, J. R., Mueller, P. W., Flückiger, W., and Rutter, A. J.: The effect of dust on photosynthesis and its significance for roadside plants, Environ. Pollut. Control, 34, 171-190, doi: 10.1016/0143-1471(84)90056-4, 1984.

Line 97: Hirano, T., Kiyota, M., and Aiga, I.: Physical effects of dust on leaf physiology of cucumber and kidney bean plants, Environ. Pollut., 89, 255-261, doi: 10.1016/0269-7491(94)00075-O, 1995.

EXPERIMENTAL PROCEDURE

**R1.2**

The methodology section describes in quite some details the novel experimental protocol implemented in this research. However, some elements could be further clarified. Additionally, a figure (or photo) representing the entire setup, including the device to spread the ash and the imaging system, would help the reader to understand the protocol.

We thank the reviewer for his/her suggestion. A new figure (Fig. S4) showing the experimental setup is provided as Supplementary materials.

**R1.3**

- Line 111: specify the typical height of plants and typical surface areas of leaves at the time of experiments;

Information on the height of tomato and chilli pepper plants before ash application is now provided.

Estimating the surface area of individual leaves is not straightforward as it varies with the leaf position on the stem. Instead, we provide the surface area of tomato's and chilli pepper's foliage as estimated by performing image analysis (using ImageJ) of the plant photos.

*Line 121: They were exposed to ash six weeks after sowing, when tomato and chilli pepper plants were at the seven- and eight-leaf stage, respectively. The corresponding plant heights were ~40 and ~30 cm. The foliage surface area was ~400 and ~100 cm² for tomato and chilli pepper, respectively;*

**R1.4**

- Line 119: crushing phonolite rocks: did you check the morphology of the produced particles using SEM after crushing and compared to actual ash? Although I understand that crushing might be the best way to generate large ash fraction, the morphology of particles might influence their interaction with leaves.

The reviewer makes a good point. Using crushed rock as a surrogate for ash from explosive eruptions allowed us to carry out multiple tests while reducing the number of uncontrolled variables. We have examined the phonolite powder obtained after crushing (i.e. the ash surrogate) by scanning electron microscopy (SEM). We now provide SEM images of this material and for each size fraction (≤90, 90-125, 125-250, 250-500, 500-1000, 1000-2000 µm) (Fig. S1 in the Supplementary materials). Regardless of the ash grain size, most particles are blocky, but rounded or platy shapes also occur. Similar shape characteristics are commonly reported for ash particles from explosive eruptions (Wohletz, 1983; Wohletz and Heiken, 1992; Coltelli et al., 2008; Nurfiani and Bouvet de Maisonneuve, 2017). Blocky particles usually result from fragmentation and quenching of magma, whereas elongated and rounded shapes are related to ductile deformation. Vesicular particles (concave and with an irregular shape) are also a typical product of explosive eruptions; they are formed by the

rupturing of vesiculated magma in the volcanic conduit. However, such particles cannot be generated by crushing a solid rock and therefore, they were absent in our ash material. While we acknowledge that this is a limitation of our experimental approach, we argue that it does not jeopardise the validity of our observations as ash deposits from explosive eruptions always contain a mixture of particle shapes. It may be speculated that the irregular shape of vesicular ash could favour its retention on leaf surfaces, in particular when these are pubescent (hairy) or wet. Thus, for a given ash mass load, the presence of vesicular particles could lead to higher retention compared to what we measured experimentally. Our results may then be regarded as conservative estimates.

We have added new text to the Materials and methods (Lines 134-141) and Discussion (Lines *307-312*) sections, where we present briefly the shape characteristics of the ash material used in our study and discuss the potential limitations, respectively.

*Lines 134-141: The shape characteristics of the six ash size fractions obtained by grinding the Laacher See phonolite were examined by scanning electron microscopy (SEM). The SEM images (Fig. S2) reveal that, regardless of their size, most particles are blocky, but rounded and platy shapes also occur. Similar shapes are commonly reported for ash particles from explosive eruptions (e.g., Wohletz (1983); Coltelli et al. (2008); Nurfiani and Bouvet de Maisonneuve (2017)). However, the vesicular ash type that is also often associated with the fragmentation of gas-rich magmas cannot be generated by rock grinding and was absent in our experimental ash material.*

*Lines 307-312: As mentioned above (section Material and methods), an inherent limitation of our experimental study is that the ash material did not contain the vesicular particles that are usually found in various proportions in ash fallout from explosive eruptions. We speculate that the irregular shape of vesicular ash could enhance retention on foliage, perhaps even more so if the leaf surfaces are pubescent or wet. Thus, our measurements may be regarded as conservative estimates.*

**R1.5**

- Ash loading of 570 g m-2 (line 129): why did you select such loading that could be considered very low.

Our research objective was twofold: (i) to assess how ash grain size, leaf pubescence and humidity conditions at leaf surfaces influence the retention of ash in tomato and chilli pepper plants, and (ii) to use this information to propose a conceptual model for predicting crop yield loss when ash does not threaten plant integrity. We conducted trials and noticed that for ash loads ≥ 1000 g m$^{-2}$ (equivalent to a deposit thickness of ~1 mm, assuming a deposit bulk density of 1 g cm$^{-3}$) tomato and chilli pepper plants were affected by lodging. Obviously, this phenomenon had to be avoided as much as possible. Therefore, we tested lower ash mass loads and found that with ~570 g m$^{-2}$ (~0.6 mm), we could replicate the experiments satisfactorily. If this ash load seems low, it represents well a situation commonly encountered at distal sites (and therefore, across a wider area) affected by ash fallout. Until our study, there were no data on the potential impact on crops of such low ash mass loads/thin deposits. Thus, our measurements contribute to fill this gap in knowledge.

We have added the following sentence to the *Material and methods* section:

*Lines 148-159: An ash load of ~570 g m$^{-2}$ was selected for the experiments. Assuming a bulk density of 1 g cm$^{-3}$ for the ash deposit (Eychenne et al., 2012), this corresponds to a relatively thin deposit of ~0.6 mm (i.e. considering a bulk deposit density of 1 g cm$^{-3}$, Eychenne et al. (2012)), best representing accumulations encountered at distal sites (and over wide areas) affected by ash fallout from explosive eruptions (Fierstein and Nathenson, 1992; Jenkins et al., 2022). Pre-tests carried out with higher ash loads (≥ 1000 g m$^{-2}$) already led to lodging of some tomato and chilli pepper plant specimens, a phenomenon that needed to be avoided in order to maximise the experiment's reproducibility. Neild et al. (1998) and Craig (2015) consider that an ash mass load of 6-30 kg m$^{-2}$ on plants leads to mechanical damage. Our observations indicate that lower loads can affect crop plants. In other words, the threshold value above which mechanical injury occurs varies with plant phenology (i.e. the combination of genotype and environment).*

**R1.6**

- Could it be expected that at larger loading, larger surface of leaves be covered until the grainsize does not matter anymore (see main comment below). Further discussion on the potential impact of ash loading on the observed relationship would be useful.

The reviewer makes an interesting comment. We contend that for intermediate ash grain sizes (90-500 µm), greater ash mass loads may lead to higher ash retention on plant foliage. More particles may accumulate along the leaf primary and secondary veins as these less elastic tissues are able to absorb the kinetic energy of impinging ash particles with an intermediate grain size. However, we believe that a higher loading of ash ≤90 µm will not lead to higher retention because a portion of the leaf surface area (~10% for tomato and chilli pepper in our experimental conditions as estimated from the data shown in Fig. 1) is too vertical to retain the ash particles. Due to their high kinetic energy, coarse particles (≥500 µm) tend to lodge on less elastic structures such as leaf folds. We predict that the retention of coarse ash on foliage will be limited by the number of leaf folds and thus, will probably not increase significantly for higher ash load values. Based on these arguments, we anticipate that, for ash loadings >570 g m$^{-2}$, the exponential relationship between ash grain size and ash retention will no longer hold and instead, a linear function would provide a better model. The fine and coarse fractions would then correspond to the maximum and minimum retention values, respectively.

We have added new text in the *Discussion* section:

*Lines 269-280: This relationship was established for a single ash mass load (~570 g m$^{-2}$). For ash in the intermediate size range, a higher load could result in enhanced retention of the particles, particularly along the primary and secondary leaf veins as these consist of less elastic tissues that can better absorb the kinetic energy of impinging ash particles of intermediate grain size. However, for fine ash, we do not expect more retention to occur if tomato and chilli pepper leaves were exposed to higher loads because a large proportion of the foliage is comprised of leaves that, due to their steep angle, cannot retain ash particles efficiently. As mentioned earlier, coarse ash particles tend to lodge primarily on leaf folds. Thus, their retention on foliage will likely be limited by the number of leaf folds. Overall, we anticipate that for ash load values >570 g m$^{-2}$, the exponential dependence of ash retention on ash grain size will start to degrade and instead, a linear relationship would be a better model.*

**R1.7**

- Line 133: "through a 2 cm-mesh sieve" – is this correct? 2 cm seems extremely coarse relative to the ash used and would not help to distribute the ash evenly across the device.

Yes, it is correct. The mesh sizes of the sieve sitting on top of the PVC tube and of the three sieves installed inside it are 2 and 1 cm, respectively. This set up allowed the formation of a uniform ash deposit on the ground.

The text was slightly modified as follows:

*Line 162-166: The ash fractions <1000 µm were poured carefully through a 2 cm-mesh sieve installed on the top of the PVC tube. Bouncing of the ash particles passing through the three inner 1-cm sieves allowed formation of a uniform deposit. Application of the coarsest ash (1000-2000 µm) was carried out with the same device, but the inner meshes were removed.*

**R1.8**

- Lines 135 and following: the protocols should be more specific for what concern the timing and location of different actions. What was the duration between the spraying of water on the leaves (for wet condition), the spreading of the ash and the acquisition of the pictures? Did these action follow each other within minutes or were there hours/days in between? Was spreading of ash conducted at the same location of the acquisition of the picture or was the plant displaced?

We thank the reviewer for his/her suggestion. We created a new figure (Fig. S4) in the Supplementary materials which indicates the timing and location of each step performed in order to collect the experimental data.

The main text was modified as follows:

*Lines 170-172: Water spraying of the plant foliage, ash application and photo acquisition all took place within the black chamber. Less than five minutes elapsed between the spraying operation and photo acquisition of the ash-treated plant (Fig. S4).*

**R1.9**

- Line 145: precise here that image is acquire before and after ash fallout (mentioned later on, but needed here for better understanding).

We have revised the text as follows:

*Lines 183-185: We analysed the digital photos taken just before and after ash application with ImageJ 1.52 (Schindelin et al., 2015). The foliar cover, a measure of the vertical projection of exposed leaf area, was estimated using a dedicated macro (https://github.com/NoaLigot/ImageJ-macro.git).*

**R1.10**

- Ash retention: was there any way to quantify the proportion of ash that was retained on the leave versus the ash that reached the ground? Were the plant weighted before and after the ash fallout?

In our experiment, the mass of ash loaded into the PVC tube is higher than that collected by tomato and chilli pepper's foliage. This simply reflects the fact that an unknown amount of the ash applied is not retained by leaves. However, using the proportion of ash found on leaves when referring to ash load would not be useful for developing models of impacts. The reason is that the metric used for describing the ash hazard intensity in risk analysis is ash accumulation (i.e. mass load or thickness, Wilson and Kaye, 2005; Jenkins et al., 2015) on the ground (and not on plant foliage), as measured directly in the field or estimated from ash dispersion/fall models. In our study, the ash hazard intensity would correspond to an ash mass load of ~570 g $m^{-2}$. Knowledge of the precise amount of ash retained on crop leaves would be needed for developing a detailed process-based understanding of impacts, but this falls out of the scope of our research. Here, we dismiss the reviewer's comment.

**R1.11**

- Lines 160-165: issue of leave bending. Authors report that some of their measurement returned higher 'green leaf surface' after ash exposure than before, claiming that this is due to movement of leaves and camera during image acquisition. As these issues probably affected all their measurement, the accuracy of the documented covered leave surface could be derived by considering the noise observed for experiments were the retention is close to zero. Additionally the issue of leave bending should receive further attention in the description of results: did significant bending or change in orientation of leaves were observed? For which grain size? Beyond the impact on the imaging procedure, the bending would also directly influence the potential of retention of ash? This is mentioned on line 266 ('which pulls a leaf downward') but no comment is made on whether this process was observed during experiments;

We appreciate the reviewer's comment. The errors linked to camera positioning and image analysis were systematic. In order to minimise these, constant light conditions were used and the results of the image analysis were validated based on a comparison to the RGB photos. Leaf bending occurred only when fine ash (<90 μm) was applied to tomato and chilli pepper; it did not affect the plants in any of the other ash treatments. Plants were tutored in order to limit stem and leaves movements. The variability in the measurements was similar across the ≤90 and ≥500 μm ash size fractions tested (Fig. 1). Therefore, we argue that leaf bending had a negligible impact on the results. In the original submission, we probably drew too much attention on the potential influence of leaf bending on ash retention. We have rectified this by removing Lines 160-165 from the original manuscript. Moreover, since we cannot disentangle errors related to camera positioning and image treatment from the "natural" data dispersion, we have modified Figs. 1, 2 and 6, Tables S1 and S2 and Fig S5 in order to display the actual data variability rather than omitting the negative values (as it was done in the original submission).

OTHER FACTORS

**R1.12**

- Ash loading: authors decide to work with a single ash loading for all experiments. They properly argue that they select an ash loading that is below the threshold for physical damage for the plant (is such threshold well defined? Is it plant specific?).

Here, the reviewer points to a deficiency in knowledge. The ash mass load threshold above which crop plants undergo mechanical damage is poorly constrained. It is plant-specific since it depends on plant's phenotype (i.e. the combination of genotype and environment). Previous field-based studies, i.e. Neild et al. (1998); Craig et al. (2021), suggest ash deposit values on the order of ten kilograms per square meter, e.g., ~30 kg $m^{-2}$ (assuming a deposit bulk density of 1 g $cm^{-3}$) for fruiting crops such as tomato and chilli pepper and 6-25 kg $m^{-2}$ for horticultural plants. However, in our experiment,

tomato and chilli pepper at the seven- and eight-leaf stage, respectively, were already affected by lodging when exposed to much less ash, i.e. ~0.6 kg m$^{-2}$. New studies are needed to better characterise the exposure conditions under which a crop plant may suffer from mechanical damage.

We have clarified this in the Material and methods sections as follows:

Lines *148-159: An ash load of ~570 g m$^{-2}$ was selected for the experiments. Assuming a bulk density of 1 g cm$^{-3}$ for the ash deposit (Eychenne et al., 2012), this corresponds to a relatively thin deposit of ~0.6 mm (i.e. considering a bulk deposit density of 1 g cm$^{-3}$, Eychenne et al. (2012)), best representing accumulations encountered at distal sites (and over wide areas) affected by ash fallout from explosive eruptions (Fierstein and Nathenson, 1992; Jenkins et al., 2022). Pre-tests carried out with higher ash loads (≥ 1000 g m$^{-2}$) already led to lodging of some tomato and chilli pepper plant specimens, a phenomenon that needed to be avoided in order to maximise the experiment's reproducibility. Neild et al. (1998) and Craig (2015) consider that an ash mass load of 6-30 kg m$^{-2}$ on plants leads to mechanical damage. Our observations indicate that lower loads can affect crop plants. In other words, the threshold value above which mechanical injury occurs varies with plant phenology (i.e. the combination of genotype and environment).*

**R1.13**

Assumption is made that the relationship between grainsize and foliage cover found for this ash loading would be valid also for other loading (or at least the type of relationship – lines 231-32). Would the retention of ash not relatively increase with increasing ash loading? Until a point were all the leave surface are covered irrespective of grainsize?

This comment duplicates a remark made earlier by the reviewer (R1.6). We kindly refer the Editor to our reply above.

Could it be assumed that once a first layer of ash is retained on the vegetation, the effect of grainsize on accumulation would not be valid, the ash particles creating their own roughness at the surface? Further discussion on the ash loading for which the observed role of grainsize might be valid should be further discussed.

For a natural ash deposit, with particles varying widely in size, the formation of a fine ash deposit at the leaf surfaces could facilitate subsequent accumulation of coarser ash if the deposit is able to absorb the kinetic energy of the impinging particles. For smooth leaf plants such as chilli pepper, a such process would increase leaf surface roughness. In order to test this, new experiments in which ash retention is quantified for a material with a broad grain size distribution would be needed.

Similarly the reader should be reminded that the yield loss mentioned are only valid for the ash loading used in the experiment and that ash loading will most probably be a significant parameter in controlling foliage cover.

We agree with the reviewer's suggestion. We have revised the text accordingly:

Line 345: *Our experimental results indicate that ~570 g m$^{-2}$ of fine ash can readily cover the upper side of leaves (Fig. 2).*

Line 396: *To illustrate our approach, we estimated CYL_% for tomato and chilli pepper plants exposed to ~0.6 mm (~570 g m$^{-2}$) of ash.*

Line 542: *We also showed that, for a given ash mass load (~570 g m$^{-2}$), the leaf surface percentage covered by ash is an exponential decay function of grain size of which the parameters are influenced by leaf pubescence and humidity conditions at leaf surfaces.*

**R1.14**

- Residence time of ash: very limited attention is giving to the time component; Authors consider the timing of the ash fallout relative to the growth of the plant, but not the duration of the ash retention on the leave (assuming early senescence of ash covered leaves). As the duration of residence not been considered in previous study? For how long does the ash need to cover the leave to cause decay? In intro (line 87-88) and discussion (line 449-450) this issue of duration should be shortly mentioned (in relation to wind/rain 'erosion')

While the reviewer's comment is valid, it does not apply directly to our study. In natural conditions, an ash deposit on leaves can be remobilised by the action of the wind and/or rain. The time necessary for eliciting leaf damage has never been studied, but undoubtedly varies depending on the phenotype of the plant. More information (which could be acquired in controlled experiments) on the timing of ash

fallout and subsequent impact on crop foliage would be needed in order to factor residence time in our model of potential yield loss. The importance of ash residence time on crop foliage in dictating impacts was already mentioned in the original submission:

Lines 521-527 (Discussion): *In addition, in the natural environment, wind- and rain-driven erosion processes can remove ash deposited on foliage. Conversely, light rain may induce crusting of ash, prolonging its residence time on leaves (Miller, 1966; Ayris and Delmelle, 2012; Le Pennec et al., 2012; Ligot et al., 2022). The significance of these environmental variables in controlling ash retention time by leaves has never been assessed quantitatively, calling for further field and experimental investigations linking ash residence time on plants and impact.*

**R1.15**

- Physical integrity: authors systematically mention that they consider impact of ash on foliage for loading below the loading required to affect 'plant integrity' (line 99). However, this threshold is not clearly defined (line 304: 'cm-thick'). I guess this threshold will be specific for each plant and development stage of a plant. This could be further clarified in discussion.

This comment is similar to R1.12. We kindly refer the editor to our reply above.

IMPLICATIONS

**R1.16**

In both the introduction (lines74-75) and conclusion (line 491), authors claim that understanding and quantifying the retention of ash on crop foliage represent an essential step in mitigating the impact of eruption on agriculture. I agree that the presented results will contribute to better assess quantitatively the potential impact of ash fallout on crops (reduced yield), however it is unclear to me what the author consider as potential mitigation measures that could be derived from these results. The mitigation actions should be specified or the focus should be on the impact assessment.

The reviewer makes a good point. The wording "mitigation measures" is not appropriate. Our experimental data allowed us to propose a model to predict potential crop yield loss based on two ash fall intensity metrics (i.e. mass load and grain size). As such, our study does not provide direct insights into the mitigation measures to be implemented by farmers after ash fallout. Nevertheless, we argue that controlled experiments allow for the production of unique datasets that can inform decision making, for instance, in relation to aid allocation, land-use planning, or parametric insuring.

The text was adjusted accordingly (Lines 75-78 and 562-565):

Lines 75-78: *These limitations are hindering the development of accurate process-based risk assessment models that can inform targeted strategies to build resilience of agriculture-based community in the case of an explosive eruption; for example, in relation to aid allocation, land-use planning and insuring.*

Lines 562-565: *Acquiring this knowledge will significantly enhance our capacity to estimate ash-related risks to crops accurately. Governments and payout agencies need such assessments in order to develop and implement effective risk reduction strategies for ashfall damage to crops in volcanically active agricultural regions.*

SMALL EDITS

- Abstract is well written but could be shortened both in the problem statement and the results implication

We agree with the reviewer. We have removed three sentences from the abstract to make it shorter.

- Line 41: 'farming activities ARE exposed'

Corrected

- Line 48: 'economic loss' – in country with subsistence farming the issue of food shortage would also have to be considered.

The text has been revised as follows:

Lines 46-49: *As a result, crop fields impacted by ash fall produce lower or poor-quality harvests that can translate into significant economic losses to farmers and food shortages at the local or even*

*regional scale, and even more so when subsistence agriculture dominates (Neild et al., 1998; Wilson et al., 2007; Ligot et al., 2022).*

- Line 76-79: which ash thickness/loading is considered to calculate these areas of crop affected?

The text has been revised as follows:

Lines 79-81: *Jenkins et al. (2022) estimated that an explosive eruption of VEI 4 (Volcanic Explosivity Index (Newhall and Self, 1982)) on the island of Java, Indonesia, has on average a 50% probability of affecting ~700 km² of crops with 5 kg m⁻² of ash.*

- Figure 1: specify the number of experiments represented by each boxplot (is it 15?). Explanation of how to read the chamber plot (median, 25-75th quantile) should be added to caption.

The caption of Fig. 1 has been modified to include the required information:

Lines 220-222 *Each boxplot represents 15 repetitions. The median value sits within the box and represents the centre of the data. Fifty % of the data values lie above the median and 50% lie below the median.*

- Figure 3: add scale bar or specify the area imaged in the caption.

The surface area of the image (~800 cm²) is mentioned in the caption of Fig.3.

- Line 285: figure 1 highlight that surface wetness has more influence on retention for chilli pepper than tomato plan. This observation should be discussed here: I guess that leave pubescence and wetness act in a similarly way, so that wetness induces lower additional retention with tomato plants.

Here, the reviewer evokes a complex topic. A detailed discussion on the relative importance of leaf hairiness and wetness in favouring ash retention would first require a precise description of leaf surface characteristics (e.g., hydrophobicity, leaf hair density, and 3D characterisation). Such data are plant/plant variety-specific, and are not available for the tomato and chilli pepper species used in our experiment. More experiments (at various humidity conditions at leaf surfaces) would also be needed to identify the main controlling factors. The reviewer's request falls out of the scope of our study, and at this stage cannot be addressed.

- Line 320: explain what is the 'harvest index'.

A definition of the harvest index is included in the revised manuscript:

Lines 379-380: *[…] harvest index, i.e. the fraction of the total aboveground biomass allocated to the harvested parts of the plant*

- Line 335-340: explain here how the impact of ash on the plant growth is simulated through leave senescence followed by new leave growth.

We have clarified this point in the revised manuscript:

Lines 383-386: *We consider two effects of ash on plant yield: reduction in LAI and premature biomass senescence. The former leads to lower accumulated biomass after formation of the ash deposit, whereas the latter is responsible for a loss of biomass that accumulated prior to ash fall.*

We also describe in more details the temporal dynamics of *LAI*:

Lines 407-411: *On the day of the eruption, the LAI is reduced by an amount corresponding to the percentage of foliar cover coated with ash. On the following days, it re-increases as new leaves formation resumes at a rate similar to that before exposure to ash. If time permits, the LAI may reach a value identical to that of a plant that would not have received ash.*

- Figure 6: provide also the results for chilli pepper in the main text, these are important results.

The experimental results obtained for chilli pepper have been added to Fig. 6.

**References**

Coltelli, M., Miraglia, L., and Scollo, S.: Characterization of shape and terminal velocity of tephra particles erupted during the 2002 eruption of Etna volcano, Italy, Bull. Volcanol., 70, 1103-1112, doi: 10.1007/s00445-007-0192-8, 2008.

Craig, H., Wilson, T., Magill, C., Stewart, C., and Wild, A. J.: Agriculture and forestry impact assessment for tephra fall hazard: fragility function development and New Zealand scenario application, Volcanica, 4, 345 - 367, doi: 10.30909/vol.04.02.345367, 2021.

Fierstein, J. and Nathenson, M.: Another look at the calculation of fallout tephra volumes, Bull. Volcanol., 54, 156-167, doi: 10.1007/BF00278005, 1992.

Jenkins, S. F., Wilson, T. M., Magill, C. R., Miller, V., Stewart, C., W., M., and Boulton, M.: Volcanic ash fall hazard and risk: technical background paper for the UNISDR Global Assessment Report on Disaster Risk Reduction 2015, Global Volcano Model and IAVCEI, 43 pp., 2015.

Jenkins, S. F., Biass, S., Williams, G. T., Hayes, J. L., Tennant, E., Yang, Q., Burgos, V., Meredith, E. S., Lerner, G. A., Syarifuddin, M., and Verolino, A.: Evaluating and ranking Southeast Asia's exposure to explosive volcanic hazards, Nat. Hazards Earth Syst. Sci, 22, 1233-1265, doi: 10.5194/nhess-22-1233-2022, 2022.

Ligot, N., Guevara C, A., and Delmelle, P.: Drivers of crop impacts from tephra fallout: insights from interviews with farming communities around Tungurahua volcano, Ecuador, Volcanica, 5, 163-181, doi: 10.30909/vol.05.01.163181, 2022.

Neild, J., O'Flaherty, P., Hedley, P., Underwood, R., Johnston, D., Christenson, B., and Brown, P.: Impact of a volcanic eruption on agriculture and forestry in New Zealand, Ministry of Agriculture and Forestry, New Zealand, 99/2, 88 pp., 1998.

Newhall, C. G. and Self, S.: The volcanic explosivity index (VEI): an estimate of explosive magnitude for historical volcanism, J. Geophys. Res., 87, 1231-1238, doi: 10.1029/JC087iC02p01231, 1982.

Nurfiani, D. and Bouvet de Maisonneuve, C.: Furthering the investigation of eruption styles through quantitative shape analyses of volcanic ash particles, J. Volcanol. Geotherm. Res., 354, doi: 10.1016/j.jvolgeores.2017.12.001, 2017.

Wilson, T. M. and Kaye, G. D.: Agricultural fragility estimates for volcanic ash fall hazards, Institute of Geological and Nuclear Sciences, New-Zealand 51 pp., 2007.

Wilson, T. M., Kaye, G., Stewart, C., and Cole, J.: Impacts of the 2006 eruption of Merapi volcano, Indonesia, on agriculture and infrastructure, Institute of Geological and Nuclear Sciences, New Zealand, 64 pp., 2007.

Wohletz, K. and Heiken, G.: Volcanic ash, University of California Press, Berkeley, 246 pp., 1992.

Wohletz, K. H.: Mechanisms of hydrovolcanic pyroclast formation: grain-size, scanning electron microscopy, and experimental studies, J. Volcanol. Geotherm. Res., 17, 31-63, doi: doi.org/10.1016/0377-0273(83)90061-6, 1983.

**Reviewer 2's comments**

Abstract

Line 15 and throughout: 'ash fall' is commonly one word 'ashfall'

Corrected

Line 33: 'mean' should be 'means'

Corrected

Introduction

Line 40: 'is' should be 'are'

Corrected

Line 40: Define 'short-term'

The term "Short-term" is now defined:

Lines 38-41: *However, farming activities in these regions are exposed to short-term (i.e. usually less than one year) negative impacts of volcanic eruptions, an issue amplified by the expanding population living under volcanic risk.*

Line 44: True in areas where cropping farming dominates (e.g., Indonesia) but not in other countries where pastoral farming of livestock dominates

The sentence has been modified as follows:

Lines 41-43: *Where cropping activity dominates (for example, in Indonesia), widespread damage to agriculture during eruptive activity arises from crop exposure to ashfall.*

Line 48: Also food security issues in areas where farming is subsistence

The sentence has been modified as follows:

Lines 46-49: As a result, crop fields impacted by ash fall produce lower or poor-quality harvests that can translate into significant economic losses to farmers and food shortages at the local or even regional scale, and even more so when subsistence agriculture dominates *(Neild et al., 1998; Wilson et al., 2007; Ligot et al., 2022).*

Line 63: Expand/give examples

Examples are now given in the text:

Lines 61-64: *In parallel, new methodologies harvesting the potential of big Earth observation data acquired from satellite-based sensors (e.g., Landsat, MODIS and Sentinel) and interpretable machine learning are being developed to complement post-EIA studies (Biass et al., 2022).*

Line 67: I think its fair to say that tropical and semi-arid areas are increasingly being considered

The sentence has been modified to include these regions:

Lines 67-69: *Firstly, they lean on limited observational data acquired in post-EIA studies. Most of these have been conducted in temperate volcanic regions, but tropical and semi-arid environments are increasingly receiving attention.*

Line 72: Is there really no impact metric? Isn't thickness/loading used in this way currently? It's not perfect but it is still indicative of likely crop damage to some extent – as you use it to eliminate the possibility of structural damage later in the manuscript

Ash thickness or mass loading is the metric commonly used to describe hazard intensity. This was clearly stated in the original manuscript (Lines 69: *Secondly, it is assumed that ground ash accumulation (thickness or ash mass load) is the principal hazard intensity metric governing impact level on crops.*). An impact metric describes the damage caused by a given type of hazard. Here the reviewer's comment is not valid.

We slightly modified the sentence to clearly differentiate hazard and impact metrics as follows:

*Line 69: Secondly, it is assumed that ground ash accumulation (thickness or ash mass load) is the principal hazard intensity metric governing impact level on crops. However, other volcanic (e.g. ash*

*grain size, surface composition) and non-volcanic factors (e.g. environmental conditions, plant traits, crop development stage) play a key role in dictating impact and vulnerability (Jenkins et al., 2015; Ligot et al., 2022). Finally, current approaches lack an impact metric that can be applied to assess crop yield loss from ashfall.*

Line 81: 'cm' to 'centimetres'

We replaced cm by kilograms per square meter throughout the text because the latter is the correct unit for ash mass load.

Line 84: Insert 'less severe' before 'disturbances'

Done

Line 84: Change 'blankets' to 'deposits'

Done

Line 85: Change 'km2' to 'square kilometres'

Done

Line 85: Insert 'structural' before 'integrity'

Done

Line 88: Is this always true? Reference? Wouldn't the depth of cover or leachable chemistry of the deposit sometimes be the mechanism of loss?

We dismiss the first part of the reviewer's comment that relates to ash deposit depth. Line 88 in the original manuscript refers explicitly to distal impacts (i.e. where thick deposits will not occur) of ashfall on vegetation.

As hinted by the reviewer, the presence of soluble compounds on ash surfaces has also been evoked to explain the deleterious effect of ash on vegetation (Miller, 1967; Cook et al., 1981; Mack, 1981; Wilson et al., 2007). The composition and abundance of such compounds on ash vary broadly (Ayris and Delmelle, 2012), depending on eruption style. If various chemical elements are always released when a fresh ash deposit is exposed to water, these are not necessarily harmful to plant leaves. In fact, through a foliar fertilisation effect, they could be beneficial (Ayris and Delmelle, 2012). Evidence of chemically-induced injuries to foliage affected by ashfall is scarce, and typically coincides with ash emissions from phreatic or phreatomagmatic eruptions (Le Guern et al., 1980; Magill et al., 2013). In this case, ash particles deposited on leaves may contain reduced sulphur minerals (elemental sulphur, pyrite) that produces sulphuric acid upon oxidation. This reaction creates highly acidic pH values locally at the leaf surface, which in turn can damage its cuticle. In the original manuscript (Line 69), we briefly mentioned ash surface composition as a factor (among others) playing a role in dictating impact.

We have revised the text thoroughly to clarify our working hypothesis (and the argument that underpins it) as follows:

Lines 89-103: *At distal sites, in the absence of structural damage to plants, the capacity of ashfall to initiate damage to crop yield hinges on the capacity of leaves coated with a thin ash deposit to operate photosynthesis and produce biomass. While the release of harmful chemical compounds from ash can cause leaf tissue injuries and affect photosynthesis, this effect, if occurring, is limited to ash emissions from phreatic and phreatomagmatic eruptions (Le Guern et al., 1980; Ayris and Delmelle, 2012). For purely magmatic explosive events, impact on crops over a wide area far from the volcano primarily relates to the shading effect exerted by the presence of solid particles on leaf surfaces, reducing light interception and decreasing photosynthetic activity (Thompson et al., 1984; Hirano et al., 1995). Thus, ash retention on foliage (i.e., the percentage of the leaf surface area covered with ash) is a critical variable for developing accurate models that can assess and predict widespread impacts on crop production from ashfall. Although ash grain size, leaf pubescence and ambient humidity have been suspected to affect ash retention on foliage, we are still lacking a (i) systematic investigation of factors controlling ash retention on foliage and (ii) quantitative impact metric reflecting crop production loss.*

Lines 477-480: *Changes in LAI and premature biomass loss in ash-affected crops are interpreted as dependent on ash retention on leaves, a process influenced by grain size, plant traits and environmental conditions (Fig. 1). Here, we exclude the possible effect of ash surface composition on ash retention.*

Lines 90-91: Evidence to support this point/reference needed

The following references have been inserted into the text:

Line 97: Thompson, J. R., Mueller, P. W., Flückiger, W., and Rutter, A. J.: The effect of dust on photosynthesis and its significance for roadside plants, Environ. Pollut. Control, 34, 171-190, doi: 10.1016/0143-1471(84)90056-4, 1984.

Line 97: Hirano, T., Kiyota, M., and Aiga, I.: Physical effects of dust on leaf physiology of cucumber and kidney bean plants, Environ. Pollut., 89, 255-261, doi: 10.1016/0269-7491(94)00075-O, 1995.

Materials and methods

Line 103: Rationale for choosing these two plant types needed

The rational for choosing tomato and chilli pepper plants were provided in the original submission (Lines 103-105: "[..] they have a similar stand in early growth period, but tomato has hairy leaves whereas chilli pepper has glabrous leaves" and Lines 26-27: "[..] two crop types commonly grown in volcanic regions").

Line 108: Clarify that the experiments took place in Belgium

We have added this information:

Line 115: *The experiment took place in Belgium.*

Line 112: Limitation that all plants the same age needs to be acknowledged. What height and leaf sizes?

The reviewer makes a good point. Our results are valid for tomato and chilli pepper plants when at the seven- and eight-leaf stage, respectively. For most plants, leaves overlap during growth as new leaves form from above. This means that youngest leaves could protect oldest leaves from falling ash particles, limiting their exposition. We have added this point to the limitations of our study at the end of the Discussion section.

*Line 504: Another assumption made to evaluate the LAI trend over time is that the entire plant canopy received the same amount of ash. Although this was verified for tomato and chilli pepper when at the seven- and eight-leaf stage, respectively, it may not be necessarily the case at a later stage of their growth if upper leaves partly shield the surfaces of leaves located below them from direct exposure to ash.*

The height of tomato and chilli pepper plants before ash application is now provided. Estimating the size of individual leaves is not straightforward as it varies with the leaf position on the stem. Instead, we provide the surface area of tomato's and chilli pepper's foliage as estimated by performing image analysis (using ImageJ) of the plants photos.

*Line 121: They were exposed to ash six weeks after sowing, when tomato and chilli pepper plants were at the seven- and eight-leaf stage, respectively. The corresponding plant heights were -40 and ~30 cm. The foliage surface area was ~400 and ~100 cm² for tomato and chilli pepper, respectively;*

Lines 119-128: What was the morphology of the particles in relation to natural ash deposits?

The reviewer's comment duplicates a remark made by Reviewer 1 (R1.4). We kindly refer the Editor to our corresponding reply.

Lines 119-128: No surface chemistry on synthetic ash material – does/does not influence ash retention and adherence?

The most abundant soluble elements from ash are usually calcium, chlorine, sodium and sulphur. They originate from the dissolution of sulphate and halide salts formed on the ash surface during gas/aerosols-ash interactions in the eruption plume. In the cases of phreatic and phreatomagmatic eruptions, water-soluble hydrothermal minerals eroded from the conduit may also be present in the ash material (Óskarsson, 1980; Christenson, 2000; Delmelle et al., 2007). As suggested by the reviewer, soluble compounds are absent in the surface of our ash material (obtained by grinding a volcanic rock). Thus, we can dismiss surface composition as a factor that could have influenced ash retention on tomato and chilli pepper plants in our experimental study. Controlled experiments could be designed to assess the effect of soluble salts on ash retention on leaf surfaces.

Line 129: Only one very thin ash deposit thickness (~0.6 mm) tested. Ash thickness effect on retention not considered.

*The reviewer is correct and is/her concern echoes that already made by reviewer 1 (R1.6). We kindly refer the Editor to our corresponding reply.*

Line 131: Where is Fig. S1 in-text reference

*We thank the reviewer for spotting this omission. Fig. S1 has become Fig. S2 in the revised manuscript. Fig. S2 is used in Line 144:*

*Line 144: The grain size distribution of the six ash size ranges was measured between 0.04 and 2000 μm by laser diffraction (Beckman Coulter LS13 320) (Fig. S2).*

Line 136: Did you immediately dose with ash after spraying?

Line 138: Were the plants moved between ashfall and photography? Or was the ash applied in the photography box?

*These two remarks were also formulated by reviewer 1 (R1.8). We kindly refer the Editor to our corresponding reply.*

Lines 138-156: How would this method scale up for use in a real-world situation? Needs to be included in discussion

*Our data acquisition method was designed specifically for our experiments, where we had full control on various parameters. It would be very difficult and impractical to scale it up in a real-world situation. We had no intention to use it in natural conditions and we did not plan for it. We dismiss the reviewer's request to include this aspect in the discussion.*

Lines 161-164: Wouldn't this limitation apply to all measurements taken in this study? Any idea of the magnitude of this impact on the results? How is this accounted for?

*The issue of how the error on our measurements has been treated is also raised by reviewer 1 (R1.11). We kindly refer the editor to our corresponding reply.*

Results

Line 173: How is leaf pubescence included in Fig. 1?

*The results obtained for tomato plant, which has pubescent leaves, are shown in Fig. 1a, whereas those for chilli pepper, which has glabrous leaves are shown in Fig. 1b. The figure caption has been clarified as follows:*

*Lines 217-222: Percentage of foliar cover coated with ash for tomato plant, i.e. which has pubescent leaves, (a) and chilli pepper plant, which has glabrous leaves (b). The percentage of foliage cover was measured for the six grain size ranges tested in dry and wet conditions at leaf surfaces. Each boxplot represents 15 repetitions. The median value sits within the box and represents the centre of the data. Fifty % of the data values lie above the median and 50% lie below the median. Measurement outliers are displayed as dots.*

Line 176: Add 'significant' before 'effect'

*Done*

Lines 183-184: Explain the points and boxplots in the caption

*The caption of Fig. 1 has been modified to include the required information as follows:*

*Lines 220-222 Each boxplot represents 15 repetitions. The median value sits within the box and represents the centre of the data. Fifty % of the data values lie above the median and 50% lie below the median. Measurement outliers are displayed as dots.*

Line 201: Change 'that ash 63 μm in diameter' to 'that ash with a median of 63 μm in diameter'

*Done*

Figure 2: Did leaf pubescence influence these curves? Was there enough data to quantify this?

*The influence of leaf pubescence is discussed in the original submission (Lines 259-276): We also note that for the ash grain size ranges 125-250 and 250-500 μm in dry conditions, coverage of tomato*

*leaves by ash was significantly greater, on average by ~30 and ~20%, respectively, compared to chilli pepper leaves.*

Lines 219-220: Link this to the function of these parts of the leaves in the discussion

Leaf folds have no particular function. They consist of different tissues, i.e. the blade and veins of expanding leaves. We do not think that more needs to be said about leaf folds.

Figure 3: Could before photos be included too? The figure needs a scale

We thank the reviewer for his/her suggestion. Photos of control plants have been added to Fig. 3. We also provide the surface area value of the processed plant images (~800 cm²) in the figure caption.

Discussion

Lines 233-236: Was this experimental or field data?

Miller (1967)'s study reports field data, whereas Johnson and Lovaas (1969) and Witherspoon and Taylor (1970)'s are experimental measurements.

We have clarified this as follows:

Lines 280-287: *The increased ash retention when grain size decreases is in accordance with the field observations of Miller (1967) after the 1963 eruption of Irazú volcano, Costa Rica, who found a higher degree of retention of the smaller particles by crop foliage (alfalfa, maize, bean, beet, cabbage, carrot, pea, pepper, potato, radish and squash). Johnson and Lovaas (1969) and Witherspoon and Taylor (1970) reached a similar conclusion after dusting various crops (i.e. alfalfa, maize, squash, soybean, sorghum, peanut and clover) with quartz powders differing in grain size (88-175 and 175-350, and 44-88 and 88-175 µm, respectively).*

Line 243: Only true if considering a homogenous ash composition

Reviewer 2 is correct. The sentence has been modified as follows:

Lines 290-293: *Ignoring aggregation processes and considering a constant particle bulk density, the coarser the particles, the larger their terminal fall velocity and thus, kinetic energy (Dellino et al., 2005; Benson, 2015), simply reflecting that mass increases with grain size.*

Line 285: Is spraying the leaves with water an accurate representation of common humid environment?

The reviewer makes a good point. In natural environments, plant leaf surface moisture mainly originates from rainwater, dew and plant guttation. Spraying the leaves with ~1.5 g of water is probably most representative of a light rain or dew, which is characterised by a relatively homogeneous deposit made of droplets with comparable sizes (Hughes and Brimblecombe, 1994; Levia et al., 2017). In contrast, guttation tends to lead to the formation of bigger droplets located on the hydathodes, i.e. pore of the leaves allowing water exudation and mostly located at the tips and margins or edges of the leaves (Singh, 2014). Since guttation is a relatively minor process responsible for surface humidity at leaf surfaces (Singh, 2016), we assume that spraying water on plant leaves can reproduce reasonably conditions found in a humid environment. We contend that a heavy rain (which we did not intend to simulate) can remobilise ash particles and erode the ash deposit. We have inserted a new sentence into the Discussion section:

Line 339-343: *Enhanced ash retention on wet leaves likely relates to the surface tension generated by water molecules present on the leaf surface (Tabor, 1977; Israelachvili, 2011). Conversely, as plant leaves are hydrophobic (Bhushan and Jung, 2006), more water on leaves, such as after a heavy or prolonged light rain, could lead to formation of large water droplets able to erode particles from the leaf surface, thereby reducing ash retention.*

Line 289: How does the density of the phonolite used compare to the density of other ash deposits?

It seems that the reviewer has misunderstood. We always refer to the ash deposit's bulk density, i.e. accounting for pore space between individual particles. The density of the phonolite is that of the dense volcanic rock (i.e. typically 2.6-2.8 g cm$^{-3}$) prior to grinding. It is not the density of the ash deposit. We did not measure the density of the ash deposit formed on the leaf surfaces. Instead, we used a bulk density value (1 g cm$^{-3}$) typical for an ash deposit made of particles in a size range similar to those tested in our study (Eychenne et al., 2012).

Line 300: Change 'Recalling' to 'Considering'

Done

Line 304 and elsewhere: Define the 'cm-thick deposit' threshold specifically

A similar suggestion was made by reviewer 1 (R1.6). We kindly refer the Editor to our corresponding reply.

Line 317: How does the Q value for Belgium compare to Q values for more commonly volcanically active countries?

Adding Q values for more commonly volcanically active countries is not required because the final result; i.e. crop yield loss, is a ratio that is not influenced by Q.

Line 320: Define 'harvest index'

We have defined the harvest index in the revised manuscript:

Lines 379-380: *[…] harvest index, i.e. the fraction of the total aboveground biomass allocated to the harvested parts of the plant*

Line 342: Evidence/reference that 'ash deposition on leaves neither halt plant growth nor production of new leaves…'

We have added two references as per the reviewer's request:

Line 407: Neild, J., O'Flaherty, P., Hedley, P., Underwood, R., Johnston, D., Christenson, B., and Brown, P.: Impact of a volcanic eruption on agriculture and forestry in New Zealand, Ministry of Agriculture and Forestry, New Zealand, 99/2, 88 pp., 1998.

Line 407: Ligot, N.: Crop vulnerability to tephra fall in volcanic regions: field, experimental and modelling approaches, Earth and Life Institute, UCLouvain, Belgium, 285 pp., 2022.

Lines 357-359: Why are these equal?

The assumption that the canopy-to-biomass ratio is equal for tomato and chilli pepper is based on previous studies, and which are cited in the original submission (Kleinhenz et al., 2006; Elia and Conversa, 2012; Poorter et al., 2015). In our model, we hypothesised that the percentage of leaf biomass covered with ash which dies is also equal for both crops. In the absence of information on this ratio, it was the simplest and most logical approach.

Three references already used in the original submission (Line 156) have been added to the text to justify why the leaf-to-canopy biomass ratio and percentage of leaf biomass covered with ash which dies are set equal for both crops.

Line 419: Kleinhenz, V., Katroschan, K.-U., Schütt, F., and Stützel, H.: Biomass accumulation and partitioning of tomato under protected cultivation in the humid tropics, Eur. J. Hort. Sci., 71, 173-182, 2006.

Line 419: Elia, A. and Conversa, G.: Agronomic and physiological responses of a tomato crop to nitrogen input, Eur. J. Agron., 40, 64-74, doi: 10.1016/j.eja.2012.02.001, 2012.

Line 419: Poorter, H., Jagodziński, A., Ruiz-Peinado, R., Kuyah, S., Luo, Y., Oleksyn, J., Usol'tsev, V., Buckley, T., Reich, P., and Sack, L.: How does biomass distribution change with size and differ among species? An analysis for 1200 plant species from five continents, New Phytol., 208, doi: 10.1111/nph.13571, 2015.

Figure 5: Show the same graphs for chilli pepper plants in this figure

The experimental results obtained for chilli pepper have been added to Fig. 6.

Lines 379-381: Why were these distributions selected?

The reason for choosing these distributions is provided in the original submission:

Lines 447: *We posited that the values taken by factors (iii) and (iv) follow a gaussian distribution (Table S5), whereas variable (i) and (ii), which are always in the range 0-1 and positive, respectively, are described by a truncated gaussian distribution.*

Line 405: Change 'in chilli pepper exposed' to 'and chilli pepper crops exposed'

Done

Line 412: Change 'mean' to 'method'

Done

Lines 414-419: More information on this is needed to demonstrate how the approach can be scaled up from a greenhouse set-up, please

The idea behind this paragraph is not to scale up from a greenhouse to the real-world situation, but rather to present a perspective arising from our study. Our greenhouse experiment allows the identification of ash grain size, leaf surface humidity and plant pubescent as important factors dictating ash retention on plant foliage, and ultimately biomass production. Ash retention by plant foliage is influenced by various and difficult-to-constrain *in-situ* factors. While our experimental study has unveiled just a few, its results suggest that ash retention could be approximated based on plant *LAI* estimates. Since *LAI* can be retrieved from several satellite-born sensors, we argue that there is a real opportunity to monitor ash impacts on crops by exploiting the high flux of Earth observation data generated in real time.

Conclusions

Lines 489-492: It is unclear how this method and its results would add anything to existing mitigation efforts. Needs further explanation on the practical ways that this data could assist in an event

Reviewer 1 made a similar comment (R1.16). We kindly refer the Editor to our corresponding reply.

**References**

Ayris, P. M. and Delmelle, P.: The immediate environmental effects of tephra emission, Bull. Volcanol., 74, 1905-1936, doi: 10.1007/s00445-012-0654-5, 2012.

Christenson, B. W.: Geochemistry of fluids associated with the 1995–1996 eruption of Mt. Ruapehu, New Zealand: signatures and processes in the magmatic-hydrothermal system, J. Volcanol. Geotherm. Res., 97, 1-30, doi: 10.1016/S0377-0273(99)00167-5, 2000.

Cook, R. J., Barron, J. C., Papendick, R. I., and Williams, G. J.: Impact on agriculture of the mount St. Helens eruptions, Science, 211, 16-22, doi: 10.1126/science.211.4477.16, 1981.

Delmelle, P., Lambert, M., Dufrêne, Y., Gerin, P., and Óskarsson, N.: Gas/aerosol–ash interaction in volcanic plumes: New insights from surface analyses of fine ash particles, Earth. Planet. Sci. Lett., 259, 159-170, doi: 10.1016/j.epsl.2007.04.052, 2007.

Elia, A. and Conversa, G.: Agronomic and physiological responses of a tomato crop to nitrogen input, Eur. J. Agron., 40, 64-74, doi: 10.1016/j.eja.2012.02.001, 2012.

Eychenne, J., Le Pennec, J.-L., Troncoso, L., Gouhier, M., and Nedelec, J.-M.: Causes and consequences of bimodal grain-size distribution of tephra fall deposited during the August 2006 Tungurahua eruption (Ecuador), Bull. Volcanol., 74, 187-205, doi: 10.1007/s00445-011-0517-5, 2012.

Hughes, R. N. and Brimblecombe, P.: Dew and guttation: formation and environmental significance, Agricultural and Forest Meteorology, 67, 173-190, doi: doi.org/10.1016/0168-1923(94)90002-7, 1994.

Johnson, J. E. and Lovaas, A. I.: Progress report on simulated fallout studies, Colorado State University, pp., 1969.

Kleinhenz, V., Katroschan, K.-U., Schütt, F., and Stützel, H.: Biomass accumulation and partitioning of tomato under protected cultivation in the humid tropics, Eur. J. Hort. Sci., 71, 173-182, 2006.

Le Guern, F., Bernard, A., and Chevrier, R. M.: Soufrière of guadeloupe 1976–1977 eruption — mass and energy transfer and volcanic health hazards, Bulletin Volcanologique, 43, 577-593, doi: 10.1007/BF02597694, 1980.

Levia, D. F., Hudson, S. A., Llorens, P., and Nanko, K.: Throughfall drop size distributions: a review and prospectus for future research, WIREs Water, 4, e1225, doi: doi.org/10.1002/wat2.1225, 2017.

Mack, R. N.: Initial effects of ashfall from mount St. Helens on vegetation in eastern Washington and adjacent Idaho, Science, 213, 537-539, doi: 10.1126/science.213.4507.537, 1981.

Magill, C., Wilson, T., and Okada, T.: Observations of tephra fall impacts from the 2011 Shinmoedake eruption, Japan, Earth Planets Space, 65, 18, doi: 10.5047/eps.2013.05.010, 2013.

Miller, C. F.: Operation ceniza-arena: The retention of fallout particles from volcan Irazu (Costa Rica) by plant and people. Part 2, Stanford Research Institute, San Francisco, California, MU-4890, 247 pp., 1967.

Óskarsson, N.: The interaction between volcanic gases and tephra: fluorine adhering to tephra of the 1970 hekla eruption, J. Volcanol. Geotherm. Res., 8, 251-266, doi: 10.1016/0377-0273(80)90107-9, 1980.

Poorter, H., Jagodziński, A., Ruiz-Peinado, R., Kuyah, S., Luo, Y., Oleksyn, J., Usol'tsev, V., Buckley, T., Reich, P., and Sack, L.: How does biomass distribution change with size and differ among species? An analysis for 1200 plant species from five continents, New Phytol., 208, doi: 10.1111/nph.13571, 2015.

Singh, S.: Chapter Three - Guttation: new insights into agricultural implications, in: Adv. Agron., edited by: Sparks, D. L., Academic Press, 97-135, doi: doi.org/10.1016/B978-0-12-802139-2.00003-2, 2014.

Singh, S.: Guttation: mechanism, momentum and modulation, The Botanical Review, 82, 149-182, doi: 10.1007/s12229-016-9165-y, 2016.

Wilson, T. M., Kaye, G., Stewart, C., and Cole, J.: Impacts of the 2006 eruption of Merapi volcano, Indonesia, on agriculture and infrastructure, Institute of Geological and Nuclear Sciences, New Zealand, 64 pp., 2007.

Witherspoon, J. P. and Taylor, F. G., Jr.: Interception and retention of a simulated fallout by agricultural plants, Health Phys., 19, 493-499, doi: 10.1097/00004032-197010000-00003, 1970.